# GDL-DS: A Benchmark for Geometric Deep Learning under Distribution Shifts

## Abstract

Geometric deep learning (GDL) has gained significant attention in various scientific fields, chiefly for its proficiency in modeling data with intricate geometric structures. Yet, very few works have delved into its capability of tackling the distribution shift problem, a prevalent challenge in many relevant applications. To bridge this gap, we propose GDL-DS, a comprehensive benchmark designed for evaluating the performance of GDL models in scenarios with distribution shifts. Our evaluation datasets cover diverse scientific domains from particle physics and materials science to biochemistry, and encapsulate a broad spectrum of distribution shifts including conditional, covariate, and concept shifts. Furthermore, we study three levels of information access from the out-of-distribution (OOD) testing data, including no OOD information, only OOD features without labels, and OOD features with a few labels. Overall, our benchmark results in 30 different experiment settings, and evaluates 3 GDL backbones and 11 learning algorithms in each setting. A thorough analysis of the evaluation results is provided, poised to illuminate insights for DGL researchers and domain practitioners who are to use DGL in their applications.

## 1 Introduction

Machine learning (ML) techniques, as a powerful and efficient approach, have been widely used in diverse scientific fields, including high energy physics (HEP) (Duarte & Vlimant, 2022), materials science (Fung et al., 2021), and drug discovery (Vamathevan et al., 2019), propelling ML4S (ML for Science) into a promising direction. In particular, geometric deep learning (GDL) is gaining much focus in scientific applications because many scientific data can be represented as point cloud data embedded in a complex geometric space. Current GDL research mainly focuses on neural network architectures design (Thomas et al., 2018; Fuchs et al., 2020; Jing et al., 2020; Schütt et al., 2021; Thölke & De Fabritiis, 2021; Liao & Smidt, 2022), capturing geometric properties (*e.g.,* invariance and equivariance properties), to learn useful representations for geometric data, and these backbones have shown to be successful in various GDL scenarios.

However, ML models in scientific applications consistently face challenges related to data distribution shifts ($\mathbb{P}_{\mathcal{S}}(X, Y) \neq \mathbb{P}_{\mathcal{T}}(X, Y)$) between the training (source) domain $\mathcal{S}$ and the testing (target) domain $\mathcal{T}$. In particular, the regime expected to have new scientific discoveries has often been less explored and thus holds limited data with labels. To apply GDL techniques to such a regime, researchers often resort to training models over labeled data from the well-explored regimes or theory-guided simulations, whose distribution may not align well with the real-world to-be-explored regime of scientific interest. In materials science, for example, the OC20 dataset (Chanussot et al., 2021) covers a broad space of catalyst surfaces and adsorbates. ML models trained over this dataset may be expected to extrapolate to new catalyst compositions such as oxide electrocatalysts (Tran et al., 2023). Additionally, in HEP, models are often trained based on simulated data and are expected to generalize to real experiments, which hold more variable conditions and may differ substantially from simulations (Liu et al., 2023).

Despite the significance, scant research has systematically explored the distribution shift challenges specific to GDL. Findings from earlier studies on CV and NLP tasks (Chang et al., 2020; Creager et al., 2021; Yao et al., 2022b) might not be directly applicable to GDL models, due to the substantially distinct model architectures.

In the context of ML4S, several studies address model generalization issues, but there are two prominent *disparities* in these works. First, previous studies are often confined to specific scientific scenarios that have different types of distribution shifts. For example, Yang et al. (2022b) concentrated exclusively on drug-related shifts such as scaffold shift, while Hoffmann et al. (2023) investigated model generalization to deal with the label-fidelity shifts in the application of materials property prediction. Due to the disparity in shift types, the findings effective for one application might be ineffectual for another.

Second, studies often assume different levels of the availability of target-domain data information. Specifically, while some studies assume some availability of the data from the target domain (Hoffmann et al., 2023), they differ on whether such data is labeled or not. On the other hand, certain investigations presume total unavailability of the target-domain data (Miao et al., 2022). These varying conditions often dictate the selection of corresponding methodologies.

To address the above disparities, this paper presents **GDL-DS**, a benchmark to evaluate GDL models' capability of dealing with various types of distribution shifts across scientific applications. Specifically, the datasets cover three scientific fields: HEP, biochemistry, and materials science, and are collected from either real experimental scenarios exhibiting distribution shifts, or simulated scenarios designed to mimic real-world distribution shifts. Plus, we leverage the inherent causality of these applications to categorize their distribution shifts into different categories: conditional shift ($\mathbb{P}_{\mathcal{S}}(X|Y) \neq \mathbb{P}_{\mathcal{T}}(X|Y)$ and $\mathbb{P}_{\mathcal{S}}(Y) = \mathbb{P}_{\mathcal{T}}(Y)$), covariate shift ($\mathbb{P}_{\mathcal{S}}(Y|X) = \mathbb{P}_{\mathcal{T}}(Y|X)$ and $\mathbb{P}_{\mathcal{S}}(X) \neq \mathbb{P}_{\mathcal{T}}(X)$), and concept shift ($\mathbb{P}_{\mathcal{S}}(Y|X) \neq \mathbb{P}_{\mathcal{T}}(Y|X)$). Furthermore, to address the disparity of assumed available out-of-distribution (OOD) information across previous works, we study three levels: no OOD information (No-Info), only OOD features without labels (O-Feature), and OOD features with a few labels (Par-Label). We evaluate representative methodologies across these three levels, specifically, OOD generalization methods for the No-Info level, domain adaptation (DA) methods for the O-Feature level, and transfer learning (TL) methods for the Par-Label level.

Our experiments operated on 3 diverse scientific domains and 6 datasets include in total 30 different settings with 10 different distribution shifts times 3 levels of OOD info, covering 3 GDL backbones and 11 learning algorithms in each setting. According to our experiments, we observe that no approach can be the best for all types of shifts, and the levels of OOD information may benefit ML models to various extents across different applications. In the meantime, our comprehensive evaluation also yields three valuable takeaways to guide the selection of practical solutions depending on the availability of OOD data:

- For the setting with some labeled OOD data, TL methods show advantages under concept shifts. This is particularly noticeable when there are significant changes in the marginal label distribution.

- For the setting with only unlabeled OOD data, DA methods show advantages when the distribution shifts happen to the features that are critical for label determination compared with other features.

- For the case without OOD information, OOD generalization methods will have some improvements if the training dataset can be properly partitioned into valid groups that reflect the shifts.

Accordingly, we recommend three steps to GDL practitioners in handling the OOD generalization issues: 1) Assess the type of data distribution shifts in your application by leveraging some domain-specific knowledge; 2) Assess the availability of collecting some labeled or unlabeled OOD data; 3) Utilize the acquired assessments to select the appropriate category of methods. Our benchmark provides practitioners with insights for making the most suitable choice.

## 2 COMPARISON WITH EXISTING BENCHMARKS ON DISTRIBUTION SHIFTS

Prior research has constructed benchmarks tailored to diverse research fields, shifts, as well as knowledge levels, as summarized in Table 1. In this section, we discuss how GDL-DS is compared to existing distribution-shift benchmarks in the following three perspectives.

**Application Domain.** Recent works have introduced benchmarks on ML methods for distribution shifts that span across application domains, including CV (Ibrahim et al., 2023), OCR (Larson et al., 2022), NLP (Yang et al., 2022a) and graph ML (Gui et al., 2022; Ding et al., 2021). Regarding ML4S, Ji et al. (2022) proposed a drug-discovery benchmark primarily centered on graph neural

Table 1: Comparison with existing benchmarks under distribution shifts from three perspectives: Application Domain, Distribution Shift, and Available OOD Info (which means if these benchmarks leveraged learning algorithms corresponding to the three OOD-Info levels). Additional comparisons can be found at Appendix A.

| Benchmark | Application Domain | Distribution Shift | | | Available OOD Info | | |
|---|---|---|---|---|---|---|---|
| | | Covariate | Conditional | Concept | No-Info | O-Feature | Par-Label |
| WILDS (Koh et al., 2021), Hendrycks & Dietterich (2018) | CV and NLP | ✔ | ✗ | ✗ | ✔ | ✗ | ✗ |
| OoD-Bench (Ye et al., 2022) | CV | ✔ | ✗ | ✔ | ✔ | ✗ | ✗ |
| WILDS 2.0 (Sagawa et al., 2022), Yu et al. (2023) | CV and NLP | ✔ | ✗ | ✗ | ✗ | ✔ | ✗ |
| Wild-Time (Yao et al., 2022a) | CV and NLP | ✔ | ✗ | ✔ | ✔ | ✗ | ✔ |
| IGLUE (Bugliarello et al., 2022) | NLP | ✔ | ✗ | ✗ | ✗ | ✗ | ✔ |
| GOOD (Gui et al., 2022), GDS (Ding et al., 2021) | Graph ML | ✔ | ✗ | ✔ | ✔ | ✗ | ✗ |
| DrugOOD (Ji et al., 2022) | ML4S (Graph ML) only Biochemistry | ✔ | ✗ | ✔ | ✔ | ✗ | ✗ |
| **GDL-DS (Ours)** | ML4S (GDL) HEP, Biochemistry and Materials Science | ✔ | ✔ | ✔ | ✔ | ✔ | ✔ |

networks. However, no works have tried to benchmark methods in numerous scientific applications let alone a focus on GDL to deal with distribution shifts like this benchmark.

**Distribution Shift**. Previous works (Koh et al., 2021; Wiles et al., 2021; Ye et al., 2022; Gui et al., 2022) explored various distribution shifts, covering diversity shift, low-data drift, correlation shift, and so on. However, many scientific application scenarios involve distribution shifts with underlying mechanisms different from those mentioned above.

**Available OOD Info.** Most previous benchmarks (Koh et al., 2021; Gui et al., 2022) focus on the setting without any OOD data. Some studies assume the availability of OOD features and benchmark DA methods (Sagawa et al., 2022; Garg et al., 2023), and others assume the availability of some OOD labels (Bugliarello et al., 2022). Since we are to understand the gain that different levels of OOD data may bring to us, our benchmark integrates these three distinct levels.

## 3 BENCHMARK DESIGN

### 3.1 DISTRIBUTION SHIFT CATEGORIES

Let $\mathcal{X}$ be the input space, $\mathcal{Y}$ be the output space, $h : \mathcal{X} \to \mathcal{Y}$ be the labeling rule. Under the OOD assumption, we have joint distribution shift, *i.e.*, $\mathbb{P}_\mathcal{S}(X, Y) \neq \mathbb{P}_\mathcal{T}(X, Y)$. We denote $f(\cdot; \Theta)$ as the GDL model with parameters $\Theta$, and $\ell : \mathcal{Y} \times \mathcal{Y} \to \mathbb{R}$ as the loss function. Our objective is to find an optimal model $f^*$ with parameters $\Theta^*$, which can generalize best to target distribution $\mathbb{P}_\mathcal{T}$:

$$\Theta^* = \arg\min_\Theta \mathbb{E}_{(X,Y)\sim\mathbb{P}_\mathcal{T}}[\ell(f(X; \Theta), Y)] \tag{1}$$

To properly categorize the datasets for better analysis, we consider the following data model. The input variable $X \in \mathcal{X}$ consists of two disjoint parts, namely the causal part $X_c$ and the independent part $X_i$, which satisfies conditional independence with $Y$ given $X_c$, *i.e.*, $X_i \perp\!\!\!\perp Y|X_c$. We denote $\to$ as the direct causal correlation between two variables in causal inference literature (Pearl et al., 2000; 2016), and define two types of data generating process, *i.e.*, $X \to Y$ and $Y \to X$. Note that our data model does not aim to cover all possible causality relationships and we acknowledge the existence of alternative correct definitions not addressed in this work. However, the proposed data model best describes the mechanisms of application scenarios to be studied in this work.

Covariate Shift, Concept Shift, and Conditional Shift are formalized as follows. We follow the well-established definitions of these shifts and further extend the definitions to our data generation setting. The initial definitions and more details about our formulations are in Appendix A.

For the data generating process $X \to Y$, we have $\mathbb{P}(X, Y) = \mathbb{P}(Y|X)\mathbb{P}(X) = \mathbb{P}(Y|X_c)\mathbb{P}(X)$ based on the aforementioned data model. We define covariate and concept shift based on the specific factor experiencing shifts between domains $\mathcal{S}$ and $\mathcal{T}$:

† Covariate Shift holds if $\mathbb{P}_\mathcal{S}(Y|X_c) = \mathbb{P}_\mathcal{T}(Y|X_c)$, and $\mathbb{P}_\mathcal{S}(X) \neq \mathbb{P}_\mathcal{T}(X)$.

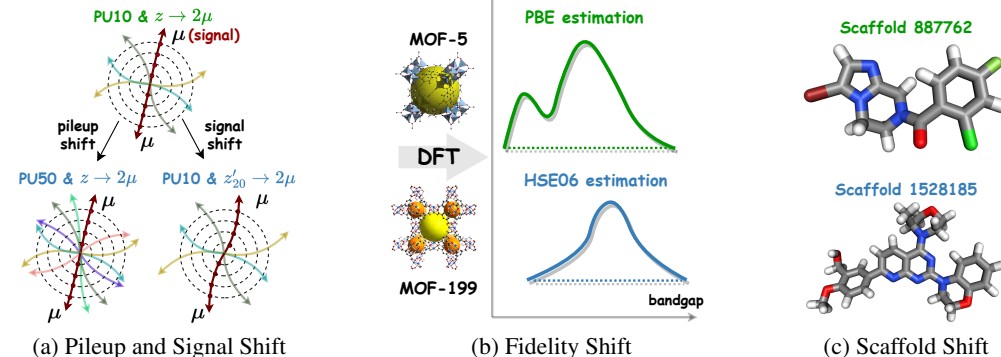

|  | (a) Pileup and Signal Shift | (b) Fidelity Shift | (c) Scaffold Shift |

Figure 1: Overview of distribution shifts in this study. The upper (green-colored) and lower (blue-colored) instances represent the scenarios in domains $\mathcal{S}$ and $\mathcal{T}$, respectively. (a) Three-dimensional trajectories of particles in a collision event, which are simulated with a magnetic field parallel to the $z$ axis and plotted on a 2D plane; (b) For the same set of MOFs, the distribution of calculated band gap values exhibits a bimodal (unimodal) nature with lower (higher) expectations under PBE (HSE06) estimation; (c) Molecular three-dimensional stick models with different scaffold IDs across $\mathcal{S}$ and $\mathcal{T}$.

Table 2: Summary of distribution shifts in this study. We also recommend applicable methods for each scenario according to our experimental results, which are shown comprehensively in Table 3.

| Scientific Field | Dataset | Domain | Shift Case | Shift Category | Evaluation Metrics | Applicable Method |
|---|---|---|---|---|---|---|
| HEP | Track-Pileup | Pileup | PU50
PU90 | $\mathcal{I}$-Conditional Shift | ACC | Mixup |
| | Track-Signal | Signal | $\tau \to 3\mu$
$z'_{10} \to 2\mu$
$z'_{20} \to 2\mu$ | $\mathcal{C}$-Conditional Shift | ACC | DANN |
| Materials Science | QMOF | Fidelity | HSE06
HSE06* | Concept Shift | MAE | TL Methods |
| Biochemistry | DrugOOD-3D-Assay | Assay | DrugOOD-lbap-core-ic50-assay | Concept Shift | AUC | GroupDRO |
| | DrugOOD-3D-Size | Size | DrugOOD-lbap-core-ic50-size | Covariate Shift | | DA or TL Methods |
| | DrugOOD-3D-Scaffold | Scaffold | DrugOOD-lbap-core-ic50-scaffold | Covariate Shift | | TL Methods |

† Concept Shift holds if $\mathbb{P}_{\mathcal{S}}(Y|X_c) \neq \mathbb{P}_{\mathcal{T}}(Y|X_c)$. Note that the shift of $\mathbb{P}(Y|X_c)$ is also characterized by the change of labeling rule $h$ between $\mathcal{S}$ and $\mathcal{T}$.

For the data generating process $Y \to X$, we have $\mathbb{P}(X, Y) = \mathbb{P}(X|Y)\mathbb{P}(Y)$. This induces the scenario of Conditional Shift, which holds if $\mathbb{P}_{\mathcal{S}}(X|Y) \neq \mathbb{P}_{\mathcal{T}}(X|Y)$ and $\mathbb{P}_{\mathcal{S}}(Y) = \mathbb{P}_{\mathcal{T}}(Y)$, and Label Shift if $\mathbb{P}_{\mathcal{S}}(X|Y) = \mathbb{P}_{\mathcal{T}}(X|Y)$ and $\mathbb{P}_{\mathcal{S}}(Y) \neq \mathbb{P}_{\mathcal{T}}(Y)$. But as label shift does not arise in our datasets, we later only focus on Conditional Shift.

Besides, we note that the conditional probability can be decomposed into two parts due to our data model, *i.e.*, $\mathbb{P}(X|Y) = \mathbb{P}(X_c|Y)\mathbb{P}(X_i|X_c)$. This enables us to further categorize conditional shifts into two sub-types based on the specific factor experiencing shifts, and we surprisingly observe that these two sub-types exhibit distinct characteristics in our experiments.

† $\mathcal{I}$-Conditional Shift holds if $\mathbb{P}_{\mathcal{S}}(X_i|X_c) \neq \mathbb{P}_{\mathcal{T}}(X_i|X_c)$, and $\mathbb{P}_{\mathcal{S}}(X_c|Y) = \mathbb{P}_{\mathcal{T}}(X_c|Y)$.

† $\mathcal{C}$-Conditional Shift holds if $\mathbb{P}_{\mathcal{S}}(X_i|X_c) = \mathbb{P}_{\mathcal{T}}(X_i|X_c)$, and $\mathbb{P}_{\mathcal{S}}(X_c|Y) \neq \mathbb{P}_{\mathcal{T}}(X_c|Y)$.

Given that each category mentioned above has only one factor experiencing shifts, we naturally partition sub-groups within the source domain $\mathcal{S}$ based on the specific factor undergoing changes.

## 3.2 DATASET CURATION AND SHIFT CREATION

In this section, we introduce the datasets in this study. Table 2 gives a summary. For each dataset, we first introduce the significance of its associated scientific application. Then, we delve into how the distribution shift of each dataset comes from in practice, and categorize the distribution shift according to the definition in Sec. 3.1. Additionally, we elaborate on the selection of domains $\mathcal{S}$ and $\mathcal{T}$, along with partitioning subgroups in the source domain $\mathcal{S}$ for our later experimental setup.

### 3.2.1 TRACK: PARTICLE TRACKING SIMULATION — HIGH ENERGY PHYSICS

**Motivations.** ML techniques have long been employed and have played a significant role in diverse applications of particle physics (Radovic et al., 2018), including particle flow reconstruction (Kieseler, 2020), jet tagging (Qu & Gouskos, 2020), and pileup mitigation (Martínez et al., 2019), *etc*. Typically, ML models rely on simulation data for training due to the scarcity of labeled data from real-world experiments. However, the intricate and time-varying nature of experimental environments often leads to distinct physical characteristics with simulated data used for training. For example, the pileup (PU) level, is defined as the number of noisy collisions around the primary collision in Large Hadron Collider experiments (Highfield, 2008). The PU level during the real deployment phase can differ from the PU level used to train the ML model.

**Dataset.** We create **Track**, a particle tracking simulation dataset, where each data sample corresponds to a collision event. It includes **Track-Pileup** and **Track-Signal** datasets. One collision event will generate numerous particles, each leaving multiple detector hits when passing through the detector. Each point in a sample represents a detector hit generated by a particle associated with a 3D coordinate. The task here is to predict the existence of a specific decay of interest (also referred to as *signal* in our work) in a given event, denoted by $Y$, like the decay of $z \rightarrow \mu\mu$. This can be formulated as a binary classification task in differentiating the detector hits left by the signal particles plus backgrounds ($X_c + X_i$) from those only left by just background particles ($X_i$). In this dataset, the causal direction $Y \rightarrow X$ is due to the fact that the detector hit patterns are determined by whether some type of collision happens. Here, sources of distribution shifts mainly come from the variation in the number of PU particles (note as Pileup shift in $\mathbb{P}(X_i|X_c)$) or the type of signal particles (note as Signal shift in $\mathbb{P}(X_c|Y)$).

**Pileup Shift — $\mathcal{I}$-Conditional Shift.** As illustrated in the bottom left instance of Fig. 1a, a higher PU level results in more background particle tracks in the collision while keeping the signal particle track the same. This mechanism is compatible with our definition of the $\mathcal{I}$-conditional shift as $\mathbb{P}_{\mathcal{S}}(X_i|X_c) \neq \mathbb{P}_{\mathcal{T}}(X_i|X_c)$ and $\mathbb{P}_{\mathcal{S}}(X_c|Y) = \mathbb{P}_{\mathcal{T}}(X_c|Y)$. We train the model on source-domain data with the PU level of 10 (PU10) and evaluate its generalizability on PU50 and PU90 target-domain data, respectively. The division of source-domain subgroups is based on the number of points present in the data entry (one collision event) as it can mimic the Pileup shift in terms of varying particle counts across different PU levels.

**Signal Shift — $\mathcal{C}$-Conditional Shift.** As depicted in the bottom right instance in Fig. 1a, we alter the physical characteristics of signal tracks by introducing signal particles with varying *momenta*, which leads to changes in the radius of signal tracks, while leaving the background particle tracks unchanged. Therefore, we categorize this shift as $\mathcal{C}$-conditional shift, as it satisfies $\mathbb{P}_{\mathcal{S}}(X_i|X_c) = \mathbb{P}_{\mathcal{T}}(X_i|X_c)$ and $\mathbb{P}_{\mathcal{S}}(X_c|Y) \neq \mathbb{P}_{\mathcal{T}}(X_c|Y)$. We train the model on source-domain data, where the positive samples consist of 5 different types of signal decays, all characterized by large signal track radii, making them easier to distinguish from background tracks. We evaluate the model on target-domain data with signal decays of $z'_{20} \rightarrow 2\mu, z'_{10} \rightarrow 2\mu, \tau \rightarrow 3\mu$, respectively, whose radii of signal tracks are smaller. We split the source $\mathcal{S}$ into 5 sub-groups, each corresponding to a specific type of signal decay.

### 3.2.2 QMOF: QUANTUM METAL-ORGANIC FRAMEWORKS — MATERIALS SCIENCE

**Motivations.** Materials property prediction plays a crucial role in discovering new materials with favorable properties (Raccuglia et al., 2016; Xie & Grossman, 2018). Training ML models using data with labels calculated from theory-grounded methods, such as DFT (Orio et al., 2009), to predict important materials properties, such as band gap, has been an emerging trend, accelerating the process of materials discovery. Among DFT methods, PBE techniques are popular for their cost-effectiveness. However, they are noted for producing low-fidelity results, particularly in the underestimation of band gaps (Zhao & Truhlar, 2009; Borlido et al., 2019). Conversely, high-fidelity methods exhibit highly accurate calculations but come at the cost of extensive computational resources, resulting in a scarcity of high-fidelity labeled data. Hence, there's a need for methods that allow ML models trained on low-fidelity data to generalize to high-fidelity prediction.

**Dataset.** We introduce the Quantum MOF (**QMOF**) (Rosen et al., 2021; 2022), a publicly available dataset comprising over 20,000 metal-organic frameworks (MOFs), coordination polymers, and their quantum-chemical properties calculated from high-throughput periodic DFT. Each point in a

sample represents an atom associated with a 3D coordinate. The task is to predict the band gap value of a given material as a regression problem that can be evaluated with MAE metrics. The dataset includes band gap values calculated by 4 different DFT methods (PBE, HLE17, HSE06*, and HSE06) ranging from low-fidelity to high-fidelity over the same set of input materials. This naturally forms the shifts across DFT methods at different fidelity levels, named fidelity shift, introduced as follows.

**Fidelity Shift — Concept Shift.** As illustrated in Fig. 1b, DFT methods at different fidelity levels tend to yield varying distributions of the band gap estimation $Y$ given the same set of input data $X$, thus reflecting the shift of $\mathbb{P}(Y|X)$ characterized by concept shift. Namely, the expected estimation band gap values tend to increase sequentially from PBE (the lowest estimation) to HLE17, HSE06*, and HSE06 (the highest estimation). We construct 2 separate shift cases: one with HSE06 as the target domain $\mathcal{T}$ and the other with HSE06* as the target domain. In both cases, the remaining three levels are used as the source domain $\mathcal{S}$, each serving as a subgroup in the source-domain splits.

### 3.2.3 DRUGOOD-3D: 3D CONFORMERS OF DRUG MOLECULES — BIOCHEMISTRY

**Motivations.** ML techniques have been applied to various biochemical scenarios, such as protein design (Anand & Achim, 2022), molecular docking (Corso et al., 2023), *etc.*, thereby catalyzing the process of drug discovery. Despite the success, one major challenge for ML-aided drug discovery is still the limited experimental data to cope with dynamic real-world pharmaceutical scenarios. Consequently, model performance easily degrades due to the underlying distribution shifts. Unpredictable public health events like COVID-19 may introduce entirely new targets from unseen domains. Besides, the assay environments, where biochemical properties are measured, may also largely diverge. These challenges related to the distribution shift spur a need for generalizable ML models to further advance drug discovery.

**Dataset.** We adapt DrugOOD (Ji et al., 2022) and propose **DrugOOD-3D**, with our main focus on the geometric structure of molecules and the GDL backbones. We leverage a conformer for each molecule and then assign a 3D coordinate to each atom. We adopt the task of Ligand Based Affinity Prediction (LBAP) in predicting the binding affinity of a given ligand molecule. We transition the task to a binary classification problem, using AUC scores for evaluation metrics, in line with the simplifications made for the regression task as also done by DrugOOD. We built **DrugOOD-3D-Scaffold**, **DrugOOD-3D-Size**, and **DrugOOD-3D-Assay** datasets, which corresponds to 3 sources of distribution shifts as follows.

**Scaffold & Size Shift — Covariate Shift.** The scaffold pattern, illustrated in Fig. 1c, is a significant structural characteristic to describe the core structure of a molecule (Yongye et al., 2012). Analogously, the molecular size is also an important biochemical characteristic. We categorize the two shifts as covariate shifts because the shift in scaffold and size primarily reflects a shift in the marginal input distribution $\mathbb{P}(X)$ across domains, while the labeling rule $h$ and $\mathbb{P}(Y|X)$ are kept invariant.

**Assay Shift — Concept Shift.** We classify assay shift as concept shift since shifts in assays can be viewed as modifying the experimental procedures and conditions. Such modifications may alter the resulting binding affinity value for the same set of molecules, described as a change in $\mathbb{P}(Y|X)$. Note that we adopt three distribution shift cases in DrugOOD, *i.e.*, lbap-core-ic50-assay, lbap-core-ic50-scaffold, and lbap-core-ic50-size, covering assay, scaffold, and size shifts, and follow the same design of domain splits and sub-group splits as DrugOOD.

## 4 EXPERIMENTS

### 4.1 EXPERIMENTAL SETTINGS

Here, we briefly introduce the experimental settings and leave more detailed descriptions of dataset splits in Appendix C, backbones and learning algorithms in Appendix D, and hyperparameter tuning in Appendix E.

**Backbones.** We include three GDL backbones widely used in scientific applications: EGNN (Satorras et al., 2021), DGCNN (Wang et al., 2019), and Point Transformer (Zhao et al., 2021).

**Learning Algorithms.** We select **11** representative methods that span a broad range of learning strategies under different levels of OOD info, i.e., No-Info, O-Feature, and Par-Label levels. For **No-**

Table 3: Experimental results (Test-ID and Test-OOD performance) on **Pileup** (PU50 and PU90 cases), **Signal** ($\tau \to 3\mu$ and $z'_{10} \to 2\mu$ cases), **Size**, **Scaffold**, and **Fidelity** (HSE06 and HSE06* cases) shifts over the backbones of EGNN and DGCNN. Note that Test-ID performance of TL methods is not evaluated. Parentheses show standard deviation across 3 replicates. ↑ denotes higher values correspond to better performance, whereas ↓ denotes lower for better. We **bold** and underline the best and the second-best OOD performance, and use † to mark best within the No-Info level for each distribution shift scenario.

**Pileup Shift — $\mathcal{I}$-Conditional Shift (ACC↑)**

| | | EGNN | | | | DGCNN | | | |
| | | PU50 | | PU90 | | PU50 | | PU90 | |
| Level | Algorithm | Test-ID | Test-OOD | Test-ID | Test-OOD | Test-ID | Test-OOD | Test-ID | Test-OOD |
|---|---|---|---|---|---|---|---|---|---|
| No-Info | ERM | 95.75(0.08) | 87.65(0.30) | 96.11(0.15) | 80.99(1.40) | 94.35(0.42) | **86.56(0.96)**† | 94.35(0.42) | 79.84(1.08) |
| | VREx | 95.49(0.32) | 87.45(0.76) | 95.49(0.32) | 80.77(0.93) | 94.54(0.17) | 86.37(0.84) | 94.54(0.17) | **80.41(0.91)**† |
| | GroupDRO | 93.18(0.33) | 83.03(0.32) | 93.18(0.33) | 75.67(0.63) | 91.48(0.19) | 79.38(0.59) | 91.48(0.19) | 73.69(0.37) |
| | DIR | 95.10(0.09) | 85.59(0.45) | 95.10(0.09) | 78.47(0.17) | 94.01(0.29) | 84.33(0.65) | 94.01(0.29) | 75.73(1.39) |
| | LRI | 95.80(0.14) | 88.15(0.31) | 95.77(0.43) | 81.43(0.80) | 93.95(0.04) | 85.25(0.04) | 93.95(0.04) | 78.65(0.39) |
| | MixUp | 95.78(0.41) | **89.41(0.11)**† | 95.86(0.13) | **82.29(0.40)**† | 94.18(0.25) | 85.93(0.46) | 94.18(0.25) | 79.43(0.77) |
| O-Feature | DANN | 95.18(0.51) | 87.16(0.72) | 95.86(0.10) | 80.69(0.44) | 93.91(0.47) | 85.01(0.64) | 94.33(0.12) | 76.15(1.95) |
| | Coral | 95.13(0.27) | 86.98(0.80) | 95.19(0.07) | 78.99(1.79) | 94.17(0.21) | 84.61(1.04) | 94.66(0.16) | 77.08(0.94) |
| Par-Label | $TL_{100}$ | | 84.20(0.46) | | 77.85(0.59) | | 81.65(1.06) | | 73.48(0.39) |
| | $TL_{500}$ | | 87.05(0.46) | | 82.09(0.88) | | 84.41(1.06) | | 78.19(0.94) |
| | $TL_{1000}$ | | 87.61(0.13) | | **83.40(0.80)** | | 85.09(0.70) | | 79.97(0.38) |

**Signal Shift — $\mathcal{C}$-Conditional Shift (ACC↑)**

| | | EGNN | | | | DGCNN | | | |
| | | $\tau \to 3\mu$ | | $z'_{10} \to 2\mu$ | | $\tau \to 3\mu$ | | $z'_{10} \to 2\mu$ | |
| Level | Algorithm | Test-ID | Test-OOD | Test-ID | Test-OOD | Test-ID | Test-OOD | Test-ID | Test-OOD |
|---|---|---|---|---|---|---|---|---|---|
| No-Info | ERM | 97.15(0.20) | 65.98(0.77) | 96.85(0.23) | 70.72(1.28) | 95.72(0.23) | 65.30(1.03) | 95.37(0.12) | 69.04(0.31) |
| | VREx | 96.86(0.29) | 66.38(0.80) | 96.66(0.15) | 71.46(0.87) | 95.66(0.03) | 64.91(0.47) | 95.49(0.32) | 69.83(0.06) |
| | GroupDRO | 96.48(0.22) | 67.15(0.10) | 96.71(0.08) | 72.56(0.88)† | 95.02(0.06) | 66.01(0.33) | 95.06(0.32) | 69.76(0.03) |
| | DIR | 77.91(2.87) | 67.32(0.43) | 93.56(2.62) | 70.04(1.00) | 91.87(0.53) | 64.74(0.82) | 91.87(0.53) | 70.67(0.81)† |
| | LRI | 96.25(0.16) | 67.49(0.24)† | 96.37(0.21) | 69.91(0.89) | 90.50(0.89) | 67.82(0.06)† | 93.40(0.28) | 67.84(0.07) |
| | MixUp | 96.95(0.22) | 65.63(0.77) | 96.97(0.06) | 71.39(1.49) | 95.41(0.33) | 66.02(1.01) | 95.80(0.08) | 69.23(0.01) |
| O-Feature | DANN | 81.37(1.04) | 68.05(0.09) | 90.08(0.86) | **77.36(0.83)** | 80.72(1.34) | **68.27(0.36)** | 87.90(0.22) | **75.46(0.53)** |
| | Coral | 96.32(0.67) | 66.61(0.49) | 96.82(0.18) | 71.60(0.42) | 94.93(0.16) | 65.06(0.43) | 94.29(0.04) | 68.95(0.73) |
| Par-Label | $TL_{100}$ | | 64.08(1.01) | | 74.21(0.94) | | 64.37(0.29) | | 63.19(0.03) |
| | $TL_{500}$ | | 67.08(0.04) | | 75.81(0.86) | | 65.99(0.76) | | 66.42(0.45) |
| | $TL_{1000}$ | | 67.47(0.11) | | 77.02(0.37) | | 65.73(0.78) | | 66.03(0.33) |

**Size & Scaffold Shift — Covariate Shift (AUC↑)**

| | | EGNN | | | | DGCNN | | | |
| | | Size | | Scaffold | | Size | | Scaffold | |
| Level | Algorithm | Test-ID | Test-OOD | Test-ID | Test-OOD | Test-ID | Test-OOD | Test-ID | Test-OOD |
|---|---|---|---|---|---|---|---|---|---|
| No-Info | ERM | 91.06(0.26) | 64.98(0.54) | 84.73(0.48) | 68.16(0.82) | 89.60(0.04) | 62.56(0.61) | 81.89(0.14) | 67.05(0.49) |
| | VREx | 91.20(0.08) | 65.01(0.50)† | 84.76(0.54) | 68.20(0.31) | 89.41(0.21) | 62.91(0.47) | 82.95(0.43) | 68.24(0.25) |
| | GroupDRO | 86.80(0.23) | 61.11(0.31) | 85.38(0.16) | 68.07(0.66) | 83.41(0.56) | 60.55(0.02) | 83.27(0.25) | 67.57(0.18) |
| | DIR | 87.24(0.84) | 64.40(0.42) | 80.59(2.34) | 67.70(1.19) | 80.43(0.50) | 62.05(0.65) | 74.49(0.37) | 67.19(0.91) |
| | LRI | 91.00(0.32) | 64.05(0.26) | 85.00(0.79) | 67.61(0.26) | 89.50(0.30) | 63.00(0.41) | 80.20(0.75) | 67.69(0.28) |
| | MixUp | 91.02(0.46) | 63.87(0.24) | 85.36(0.29) | 68.28(0.19)† | 89.45(0.19) | 63.65(0.21)† | 82.71(0.57) | 68.33(0.69)† |
| O-Feature | DANN | 91.25(0.05) | **65.45(0.45)** | 85.65(0.42) | 67.66(0.90) | 89.08(0.37) | 63.73(0.49) | 82.30(0.69) | 67.74(0.50) |
| | Coral | 91.32(0.19) | 64.77(0.49) | 85.41(0.72) | 68.61(0.48) | 89.05(0.36) | **63.87(0.48)** | 81.92(0.69) | 67.26(0.62) |
| Par-Label | $TL_{100}$ | | 64.48(0.29) | | 67.21(0.34) | | 62.50(0.21) | | 67.10(0.18) |
| | $TL_{500}$ | | 64.84(0.28) | | 68.94(0.67) | | 62.54(0.22) | | 68.15(0.13) |
| | $TL_{1000}$ | | **65.43(0.27)** | | **70.71(0.43)** | | 63.00(0.38) | | **68.79(0.11)** |

**Fidelity Shift — Concept Shift (MAE ↓)**

| | | EGNN | | | | DGCNN | | | |
| | | HSE06 | | HSE06* | | HSE06 | | HSE06* | |
| Level | Algorithm | Test-ID | Test-OOD | Test-ID | Test-OOD | Test-ID | Test-OOD | Test-ID | Test-OOD |
|---|---|---|---|---|---|---|---|---|---|
| No-Info | ERM | 0.508(0.003) | 1.099(0.095) | 0.624(0.014) | 0.556(0.007) | 0.486(0.005) | 1.082(0.030) | 0.604(0.003) | 0.547(0.007) |
| | VREx | 0.511(0.005) | 1.083(0.063) | 0.628(0.010) | **0.534(0.012)**† | 0.511(0.002) | 1.042(0.075) | 0.620(0.002) | 0.522(0.007) |
| | GroupDRO | 0.533(0.003) | 0.996(0.029)† | 0.689(0.009) | 0.546(0.002) | 0.515(0.001) | 0.977(0.021)† | 0.698(0.004) | **0.518(0.006)**† |
| O-Feature | DANN | 0.502(0.004) | 1.161(0.017) | 0.623(0.011) | 0.570(0.012) | 0.484(0.001) | 1.051(0.030) | 0.603(0.007) | 0.540(0.009) |
| | Coral | 0.504(0.004) | 1.161(0.045) | 0.623(0.005) | 0.571(0.009) | 0.488(0.003) | 1.062(0.014) | 0.605(0.007) | 0.538(0.007) |
| Par-Label | $TL_{100}$ | | 0.732(0.009) | | 0.629(0.036) | | 0.695(0.026) | | 0.603(0.009) |
| | $TL_{500}$ | | 0.638(0.008) | | 0.556(0.013) | | 0.620(0.010) | | 0.541(0.005) |
| | $TL_{1000}$ | | **0.625(0.003)** | | 0.547(0.010) | | **0.575(0.008)** | | **0.517(0.000)** |

**Info** level, we select 1) *vanilla*: ERM (Vapnik, 1999); 2) *invariant learning*: VREx (Krueger et al., 2021); 3) *data augmentation*: MixUp (Zhang et al., 2018); 4) *subgroup robust method*: GroupDRO (Sagawa et al., 2019); 5) *causal inference*: DIR (Wu et al., 2021); 6) *information bottleneck*: LRI (Miao et al., 2022). Notably, DIR is a well-known graph-based OOD baseline and LRI is a novel algorithm grounded in GDL. We refer to the above-mentioned methods as *OOD generalization methods* for simplicity. For **O-Feature** level, we select *domain-invariant methods*: 7) DANN (Ganin et al., 2016) and 8) DeepCoral (Sun & Saenko, 2016). For **Par-Label** level, we conduct *vanilla*

transfer learning plus fine-tuning with 9) 100, 10) 500, and 11) 1000 labels, which are denoted as $TL_{100}, TL_{500}, TL_{1000}$ respectively. Note that in the fine-tuning stage, the entire model parameters will be fine-tuned. Regarding the fidelity shift, we select only a subset of OOD generalization methods (VREx and GroupDRO) that are compatible with regression tasks to evaluate.

**Dataset Splits.** For each collected dataset, we first divide it into the ID dataset and the OOD dataset based on our characterization of $\mathbb{P}_S$ and $\mathbb{P}_T$. The resulting dataset in the source domain contains multiple subgroups following our split covered in Sec. 3.2, for the operation of OOD methods that rely on subgroup splits. Subsequently, the ID and OOD datasets are randomly segmented into Train-ID, Val-ID, and Test-ID, and Train-OOD, Val-OOD, and Test-OOD, respectively.

**Model Training & Evaluation.** In the No-Info setting, we train a GDL model solely on the Train-ID dataset, running the aforementioned OOD methods; In the O-Feature setting, we run DA algorithms and train the model on both Train-ID and data features of the whole OOD dataset; In the Par-Label level, we use the Train-OOD Dataset to fine-tune the model parameters that have already been pre-trained on Train-ID. Across all levels of OOD info and algorithms, we evaluate the model's ID performance using the *same* Val-ID and Test-ID datasets, and its OOD performance using the *same* Val-OOD and Test-OOD datasets, for thorough and fair analysis.

**Hyperparameter Tuning.** We tune a predefined set of hyperparameters (provided in Appendix E) and select the model with the best metric score of Val-OOD for the ultimate evaluation.

## 4.2 RESULTS ANALYSIS — GENERAL TENDENCY

Experimental results on 2 of 3 backbones are shown in Table 3. Complete results can be found in Appendix F. We begin by giving an overall comparison and discussing some general findings.

Firstly, we find that $TL_{1000}$ (fine-tuning with 1000 labels) outperforms ERM in most cases, demonstrating the effectiveness of this strategy. However, we further observe that fine-tuning can sometimes result in negative effects when the labeled OOD data is quite limited, particularly in cases involving a smaller degree of distribution shifts. Take the case of the pileup shift as an example, $TL_{100}$ under-performs ERM by a large margin. This observation is consistent with catastrophic forgetting in Kirkpatrick et al. (2017). To mitigate this issue, the strategy of surgical fine-tuning (Lee et al., 2022) is a potential solution for this problem.

Besides, consistent with previous works (Ding et al., 2021), we observe that OOD generalization methods in the No-Info level find it hard to provide significant improvement across various applications, which implies that the assumptions adopted by these methods may be kind of strong and not really match practical scenarios. Therefore, We recommend that future studies pay attention to 1) collecting some data information from the target domain $\mathcal{T}$ if possible, and 2) proposing novel OOD methods based on assumptions that better match the scientific applications. In the process of results analysis, we find multiple previous OOD works (such as in tasks of CV) shed insights to us. Therefore, we conduct comparisons with these previous findings and put details in Appendix G.

## 4.3 RESULTS ANALYSIS — INSIGHTFUL CONCLUSIONS

Besides the general observations exhibited above, our experiments also yield some intriguing conclusions that may be widely applicable. We structure this subsection by first presenting our conclusions, exemplified by representative observations and rational explanations.

• **Conclusion 1. DA methods show significant advantages with a $\mathcal{C}$-conditional shift, i.e., the shift arising in the causal component $\mathbb{P}(X_c|Y)$, as opposed to an $\mathcal{I}$-conditional shift.**

We present 2 representative observations and give explanations by analyzing the inherent causal mechanism of the shift scenarios. A great example of this conclusion comes from the signal shift, where we observe that DANN, a DA method, performs particularly well by largely outperforming ERM (↑ 6.64% in the EGNN backbone) and all OOD generalization methods without OOD info (↑ at least 4.80% in the EGNN backbone). The $TL_{1000}$ method that accesses the most labeled OOD data even cannot outperform DANN. As introduced in Sec. 3.2.1, the signal shift represents a $\mathcal{C}$-conditional shift. The access to OOD features enables the DANN model to align the latent representation of the causal part (the signal decay in a collision event) across the source and target domains, classifying unseen signal types correctly.

In contrast, DA only yields performance very close to ERM in the pileup shift. Although both the pileup and signal shift are categorized as the conditional shift, they exhibit distinct mechanisms, as mentioned in Sec. 3.2.1. Concretely, the pileup shift represents a $\mathcal{I}$-conditional shift, where the shift happens exclusively on the independent part, *i.e.* $\mathbb{P}(X_i|X_c)$. Therefore, the OOD features, unlike in the signal shift scenario, cannot provide sufficient information to guide the model's predictions in the target domain.

• **Conclusion 2. TL methods show advantages under concept shift, particularly when the shift of the marginal label distribution $\mathbb{P}(Y)$ is large.**

We illustrate this conclusion by examining two cases of the fidelity shift, where the TL strategy demonstrates contrasting results: We find this strategy performs particularly well in the case of HSE06 (even fine-tuned on a very small number of OOD labels), where it largely outperforms all other methods with the MAE score increased by at least 30%. However, it exhibits only limited improvement in the case of HSE06*. We analyze the difference by analyzing the marginal label distributions $\mathbb{P}_{\mathcal{S}}(Y)$ and $\mathbb{P}_{\mathcal{T}}(Y)$. As mentioned in Sec. 3.2.2, the fidelity shift can be characterized by a change in the labeling rule, *i.e.*, two similar inputs may be mapped to very different $Y$ values. Specifically, fidelity levels of PBE, HLE17, and HSE06* provide estimations that are closer to each other, while the HSE06 level significantly exceeds the other three. Therefore, the case of HSE06, with the HSE06 level as the target domain $\mathcal{T}$ but the other three levels as the source domain $\mathcal{S}$, yields a large difference between the marginal distributions of labels in two domains $\mathbb{P}_{\mathcal{S}}(Y) \neq \mathbb{P}_{\mathcal{T}}(Y)$. In this scenario, the OOD labels are crucial to finetune the model predictions to match the aimed distribution $\mathbb{P}_{\mathcal{T}}(Y)$. In contrast, the case of HSE06*, with the HSE06* level as $\mathcal{T}$ but the others as $\mathcal{S}$, yields closer marginal distributions of $\mathbb{P}(Y)$ between the two domains, *i.e.*, $\mathbb{P}_{\mathcal{S}}(Y) \approx \mathbb{P}_{\mathcal{T}}(Y)$. Therefore, a small amount of OOD labels tends to have a limited impact on the model performance.

Additionally, we also observe that the TL strategies cannot yield a large improvement over ERM in the assay shift, which is another scenario of Concept Shift as mentioned in Sec. 3.2.3. We analyze it also by checking the marginal label distributions $\mathbb{P}_{\mathcal{S}}(Y)$ and $\mathbb{P}_{\mathcal{T}}(Y)$. We put the results in Appendix F and detailed illustrations for the fidelity and assay shifts in Appendix H.

• **Conclusion 3. For the OOD generalization methods to learn robust representations, the more informatively the groups obtained by splitting the source domain $\mathcal{S}$ indicate the distribution shift, the better performance these methods may achieve.**

This observation is related to GroupDRO, an OOD method that is to learn robust representation across different group splits of the training domain $\mathcal{S}$. GroupDRO almost consistently outperforms ERM in all cases with the signal shift ($\tau, z'_{10}, z'_{20}$) while it largely under-performs ERM in the cases of the pileup shift (PU50, PU90). GroupDRO captures robustness by increasing the importance of subgroups with larger errors and thus highly relies on the assumption that the shift between the splits of in-domain data can to some extent reflect the distribution shift between the source $\mathcal{S}$ and the target $\mathcal{T}$. In the cases with signal shift, the way to split subgroups of the source domain aligns well with the distribution shift: Each split represents a distinct type of decay (5 types in total). By learning robust representations across these subgroups, GroupDRO yields better OOD generalization. In contrast, in the case of pileup shift, the number of points in a collision event is used as a proxy of pileup shift to achieve the group splits of the training dataset, based on the fact that the PU level is positively correlated with the number of particles. This way of subgroup splits is subjective, which is limited by the availability of data and may not fully reflect the pileup shift between domain $\mathcal{S}$ and $\mathcal{T}$.

## 5 CONCLUSION

This work systematically evaluates the performance of GDL models in the face of distribution shift challenges encountered in scientific applications. Our benchmark has 30 distinct scenarios with 10 different distribution shifts times 3 levels of available OOD information, covering 3 GDL backbones and 11 learning algorithms. Based on our comprehensive evaluation, we reveal several intriguing discoveries. In particular, our results may help select applicable solutions based on the causal mechanism behind the distribution shift and the availability of OOD information. Moreover, our benchmark encourages more realistic and rigorous evaluations of GDL used in scientific applications, and may inspire new methodological advancements for GDL to deal with distribution shifts.

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

## A  More Discussions about Categorized Distribution Shifts

Here we give a complementary discussion about the distribution shift categories that are discussed in Sec. 3.1. Concretely, we adopt the Structure Causal Model (SCM) (Pearl et al., 2000; 2016) to represent $\mathcal{I}$-conditional, $\mathcal{C}$-conditional, covariate, and concept shifts from a causal view. As illustrated in Fig. 2, five variables, including input $X$, causal part $X_c$, independent part $X_i$, domain $D$ and label $Y$, are linked by the direct causal correlation "$\rightarrow$". For some variable $A$ in the SCM, $\mathbf{Pa}(A) \rightarrow A$ denotes as the direct causal link from its parent variables $\mathbf{Pa}(A)$ to $A$. According to the causal theory (Pearl et al., 2000; 2016), there exists the correlation $\mathbf{Pa}(A) \rightarrow A$, if and only if there exists a function $f_A$, *s.t.*, $A = f_A(\mathbf{Pa}(A), \epsilon_A)$, where $\epsilon_A$ is exogenous noise satisfying $\epsilon_A \perp\!\!\!\perp \mathbf{Pa}(A)$, and we omit the exogenous noise in this study for simplification. Plus note we treat $D$ as an additional variable that exerts an influence on the other variables and thus induces a shift in the corresponding probability distribution between the domains $\mathcal{S}$ and $\mathcal{T}$.

We start with the correlation that is shared across four categories. The input variable $X$ consists of two disjoint parts $X_i$ and $X_c$, *i.e.*, $X_i \rightarrow X \leftarrow X_c$. Note that we do not further discuss potential causal dependencies between $X_i$ and $X_c$ for simplicity, although some works (Kaur et al., 2022; Chen et al., 2022) involved them. Therefore, we use a dashed arrow to represent the potential dependencies between $X_i$ and $X_c$, following Wu et al. (2021). In the meantime, we assume the independent part $X_i$ and the label variable $Y$ to be conditionally independent given $X_c$, *i.e.*, $X_i \perp\!\!\!\perp Y|X_c$, as discussed in Sec. 3.1.

In particular, the causal part $X_c$ shares the causal correlation with $Y$, represented as either $X_c \rightarrow Y$ (which is assumed by many previous works), or $Y \rightarrow X_c$ (which appears in our study), corresponding to the aforementioned data generating process $X \rightarrow Y$ and $Y \rightarrow X$. Concretely, we classify the following distribution shifts based on their distinct data generation process between $X$ and $Y$ (specifically, the correlation between $X_c$ and $Y$) as well as how the domain variable $D$ affects individual variables like $X_i, X_c$ or $Y$. Note that we follow the well-established definitions of these shifts and further extend the definitions to what we present in our work based on our assumption to better align with application scenarios we propose.

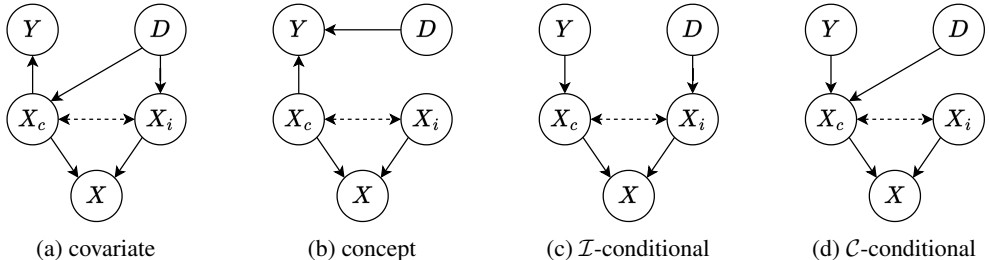

Figure 2: SCMs of covariate, concept, $\mathcal{I}$-conditional, and $\mathcal{C}$-conditional shifts.

**Covariate Shift** (Gretton et al., 2009) is initially defined as $\mathbb{P}_\mathcal{S}(X) \neq \mathbb{P}_\mathcal{T}(X)$ and $\mathbb{P}_\mathcal{S}(Y|X) = \mathbb{P}_\mathcal{T}(Y|X)$. In the context of our work, the data generating process of $X \rightarrow Y$ induces the assumption of covariate and concept shifts, and covariate shift holds if $\mathbb{P}_\mathcal{S}(X) \neq \mathbb{P}_\mathcal{T}(X)$ and $\mathbb{P}_\mathcal{S}(Y|X_c) = \mathbb{P}_\mathcal{T}(Y|X_c)$, which are achieved by the variable $D$ exclusively impacting $X$ without affecting $Y$, as shown in Fig. 2a. Note that we do not further specify which part of $X$, $X_c$ or $X_i$, is impacted by the variable $D$. This is in line with real-world scenarios where the specific shifts in $\mathbb{P}(X_c)$ or $\mathbb{P}(X_i)$ may not be apparent, such as in our instances of the scaffold shift and size shift based on the DrugOOD-3D dataset. Therefore, we present both $D \rightarrow X_c$ and $D \rightarrow X_i$ in Fig. 2a for simplicity.

**Concept Shift** (Gama et al., 2014) is initially formalized as $\mathbb{P}_\mathcal{S}(Y|X) \neq \mathbb{P}_\mathcal{T}(Y|X)$. Under our formulation, concept shift holds if $\mathbb{P}_\mathcal{S}(Y|X_c) \neq \mathbb{P}_\mathcal{T}(Y|X_c)$. This is characterized by the correlation $D \rightarrow Y \leftarrow X_c$, as shown in Fig. 2b, which means the label $Y$ is determined collectively by the input causal part $X_c$ and the domain variable $D$, and more importantly, there is a change of the labeling rule $h$ across domains $\mathcal{S}$ and $\mathcal{T}$. Note that in our study we do not further assume if the shift in $\mathbb{P}(X)$ exists. In the fidelity shift, DFT methods with varying fidelity levels calculate band gap values of the same set of MOFs, where we consider that $\mathbb{P}(X)$ remains invariant across domains, while the

assay shift, also categorized as concept shift, may involve the shift in $\mathbb{P}(X)$. So we do not explicitly present the correlation between $D$ and $X_i$ or $X_c$ in Fig. 2b.

In terms of covariate shift and concept shift, our extended formulations, compared to the initial definitions, put greater emphasis on the analysis of $\mathbb{P}(Y|X_c)$ rather than $\mathbb{P}(Y|X)$. Such analysis is crucial because underlying rationale or causal correlation, often rooted in well-established scientific rules or theories like Density Functional Theory (DFT), holds significant importance for scientific discovery and ML4S. Therefore, it deserves specific attention and emphasis.

**Conditional Shift**, as proposed in Zhang et al. (2013), is induced by the data generating process of $Y \to X$ and holds if $\mathbb{P}_\mathcal{S}(X|Y) \neq \mathbb{P}_\mathcal{T}(X|Y)$ and $\mathbb{P}_\mathcal{S}(Y) \neq \mathbb{P}_\mathcal{T}(Y)$. We follow this formulation in our work. Note that $Y \to X$ aligns well with the scenario of the Track dataset, where the simulated physical event $X$ is controlled by multiple parameters, including one representing the label $Y$ as positive or negative. The conditional distribution could be decomposed into two distinct parts based on the data model of $X_i \perp\!\!\!\perp Y|X_c$: $\mathbb{P}(X|Y) = \mathbb{P}(X_c|Y)\mathbb{P}(X_i|X_c, Y) = \mathbb{P}(X_c|Y)\mathbb{P}(X_i|X_c)$, which serves as the basis to further categorize conditional shift into the two following sub-types.

- $\mathcal{I}$-**Conditional Shift** holds if $\mathbb{P}_\mathcal{S}(X_i|X_c) \neq \mathbb{P}_\mathcal{T}(X_i|X_c)$ and $\mathbb{P}_\mathcal{S}(X_c|Y) = \mathbb{P}_\mathcal{T}(X_c|Y)$. As shown in Fig. 2c, the domain variable $D$ exclusively affects the independent part $X_i$, *i.e.*, $D \to X_i$. In this case, only the conditional distribution $\mathbb{P}(X_i|X_c)$ changes across domains $\mathcal{S}$ and $\mathcal{T}$. Note that there does not exist the causal link of $X_i \to X_c$ in this scenario to hold the assumption of $X_i \perp\!\!\!\perp Y|X_c$, so the distribution $\mathbb{P}(X_c|Y)$ will not be indirectly influenced by $D$ and thus keeps invariant across the domains.

- $\mathcal{C}$-**Conditional Shift** holds if $\mathbb{P}_\mathcal{S}(X_i|X_c) = \mathbb{P}_\mathcal{T}(X_i|X_c)$ and $\mathbb{P}_\mathcal{S}(X_c|Y) \neq \mathbb{P}_\mathcal{T}(X_c|Y)$. As shown in Fig. 2d, the domain variable $D$ exclusively affects the causal part $X_c$, which forms the structure of $Y \to X_c \leftarrow D$, representing the distribution of $X_c$ is determined by both $Y$ and $D$. That means only the conditional distribution $\mathbb{P}(X_c|Y)$ changes across the domains $\mathcal{S}$ and $\mathcal{T}$ while the distribution $\mathbb{P}(X_i|X_c)$ keeps invariant.

## A.1 ADDITIONAL COMPARISONS WITH DISTRIBUTION SHIFTS IN RELATED STUDIES

Here we conduct additional comparisons with some important distribution shifts which have been proposed by related works.

**Concept Shift.** GOOD (Gui et al., 2022) also proposed concept shift in their work, but we claim it operates under a different mechanism than the one formalized in our study. In our work, concept shift particularly denotes the change in **causal correlation** between $X_c$ and $Y$, *i.e.*, the shift in $\mathbb{P}(Y|X_c)$. This definition aligns perfectly with a real-world scenario of fidelity shift observed in materials science. In contrast, in GOOD's context, concept shift corresponds to changes in **statistical rather than causal** correlation. For instance, it may involve correlations between color and digit in datasets like GOOD-CMNIST (Gui et al., 2022).

## B PRELIMINARIES FOR GEOMETRIC DEEP LEARNING

**Notations**. We consider a geometric data sample $g = (\mathcal{V}, \mathbf{X}, \mathbf{r})$, where $\mathcal{V} = \{v_1, \cdots, v_n\}$ is a set of points with the size $n$, $\mathbf{X} \in \mathbb{R}^{n \times m}$ denotes as $m$-dimensional point features, and $\mathbf{r} \in \mathbb{R}^{n \times d}$ denotes as $d$-dimensional spacial coordinates of points. We specifically focus on 3D coordinates of scientific data in our study, *i.e.*, $d = 3$. We build the GDL model $\hat{y} = f(g; \Theta)$ to predict the ground-truth label $y$ of data $g$, where $y$ is categorical for classification tasks and continuous for regression tasks. The model in our study consists of two parts, *i.e.*, $f = \omega \circ \Phi$, including the GDL component $\Phi$, which is based on multiple GDL layers, and the MLP component $\omega$, which gives the final prediction. And we hope the GDL models maintain strong predictive performance even when $g$ is drawn from a distribution differing from the one during training, which motivates our study.

**Pipelines**. Here we present *how GDL backbones handle geometric data* in this study. Given $N$ samples of $\{g_i\}_{i=1}^N$, we begin with constructing a $k$-nn graph for each data entry based on the spacial distances, *i.e.*, $\|\mathbf{r}_v - \mathbf{r}_u\|_2$ between any pair of points $u, v \in \mathcal{V}$, where $k$ is a hyperparameter. The GDL model then iteratively updates the representation of the point $v$ via aggregation $\text{AGG}_{u \in \mathcal{N}(v)}(\mathbf{m}_{uv})$, where AGG denotes as the aggregation operator (*e.g.*, $\sum$ or max), $\mathcal{N}(v)$ denotes as the neighbors

Table 4: Dataset statistics for each dataset and distribution shift scenario. We evaluate the ID performance of models using Val-ID and Test-ID Datasets, and the OOD performance using Val-OOD and Test-OOD Datasets. The "OOD" column in this table presents the total number of OOD data entries whose features (but not labels) are used in the O-Feature level. Note that this table does not include statistics of Train-OOD that is specifically used for fine-tuning models in the Par-label level as mentioned in Sec. 4.1, because we utilize a fixed number of 100, 500, 1000 labels in this case, corresponding to $\text{TL}_{100}$, $\text{TL}_{500}$, $\text{TL}_{1000}$ baselines respectively. For bi-classification tasks, we list the number of positive data points (left) and negative data points (right), separated by "/".

| Dataset | Shift | Shift Case | Train-ID | Val-ID | Test-ID | OOD | Val-OOD | Test-OOD |
|---------|-------|-----------|----------|--------|---------|-----|---------|----------|
| Track | Pileup | PU50 | 14814/15634 | 2469/2605 | 2470/2607 | 10000/10000 | 2500/2500 | 2500/2500 |
| | | PU90 | | | | 7700/7700 | 2500/2500 | 2500/2500 |
| | Signal | $\tau \to 3\mu$ | 11851/15000 | 1975/2500 | 1975/2500 | 12000/15000 | 1975/2500 | 1977/2500 |
| | | $z'_{10} \to 2\mu$ | | | | 12000/15000 | 1975/2500 | 1977/2500 |
| | | $z'_{20} \to 2\mu$ | | | | 12000/15000 | 1975/2500 | 1977/2500 |
| QMOF | Fidelity | HSE06 | 10781 | 1796 | 1798 | 6000 | 2000 | 2000 |
| | | HSE06* | 10781 | 1796 | 1798 | 6000 | 2000 | 2000 |
| DrugOOD-3D | Assay | lbap-core-ic50-assay | 29060/3861 | 9611/1295 | 9945/1323 | 32371/4687 | 17099/1557 | 15272/3130 |
| | Size | lbap-core-ic50-size | 32686/2542 | 10872/857 | 11003/846 | 26426/6921 | 14657/2706 | 11769/4215 |
| | Scaffold | lbap-core-ic50-scaffold | 19455/1116 | 4473/211 | 26670/3015 | 30389/6824 | 16020/2678 | 14369/4146 |

of point $v$ in the $k$-nn graph, and $\mathbf{m}_{uv}$ denotes as the message passing from the point $u$ to $v$. The GDL models typically need to capture geometric properties (*e.g.*, invariance properties), and this has caused GDL models to often process geometric features carefully. Beyond basic spatial coordinates, the GDL models that achieve the invariance merit often incorporate relative geometric information between points into the message $\mathbf{m}_{uv}$, such as distance (Schütt et al., 2017), angle (Gasteiger et al., 2019), torsion (Liu et al., 2021), and rotation angle (Wang et al., 2022) information. Also note that the selected backbones in this study only involve the distance information. Investigation into how capturing higher-order geometric information like certain kinds of angles with special scientific meanings affects the generalization ability of GDL models remains a topic for future work.

After several GDL layers, there is a pooling operator used to aggregate all point representations, to obtain the representation of the geometric data. Then an additional MLP component is needed to generate the predicted labels.

## C  DETAILS OF DATASETS

### C.1  DATASET STATISTICS

The statistics of the covered datasets are shown in Table 4. Strategies of domain splits and sub-group splits for each distribution shift scenario, which have been discussed in Sec. 3.2, are detailed in Table 6. Note that for distribution shifts in DrugOOD-3D, we follow the same dataset splits and sub-group splits as the original benchmark DrugOOD. But in the Par-Label setting, we split 1000 samples from both Val-OOD and Test-OOD datasets, create the Train-OOD dataset, sample a specific number (100, 500, and 1000) for model fine-tuning, and evaluate the OOD performance of the fine-tuned models on the remaining OOD data. We also ensure a fair comparison here, as the number of removed samples is significantly smaller than the size of the OOD dataset itself. In the following three sub-sections, we make a complementary introduction to the studied scientific datasets.

Besides, we provide more granular information that better reflects the characteristics of the constructed datasets and distribution shifts. The detailed statistics can be seen in Table 5, covering

1) The average number of tracks for each pileup level in Pileup Shift (Track-Pileup Dataset). Note that a higher PU level results in more background particle tracks in the collision while keeping the signal particle track the same.

2) The average signal radius of each type of signal in Signal Shift (Track-Signal Dataset). Note that from $z \to 2\mu$, $z'_{20} \to 2\mu$, $z'_{10} \to 2\mu$, to $\tau \to 3\mu$, the average radius of signal tracks progressively

approaches 2724.96 (the average radius of background tracks), which means it is getting harder to distinguish signals from backgrounds.

3) The average number of atoms for different domains in Size Shift (DrugOOD-3D-Size Dataset).

4) The average band gap value for each fidelity level in Fidelity Shift (QMOF Dataset). Note that the distinction between these fidelity levels extends beyond the mean of bandgap values. Specifically, the distribution of calculated band gap values displays varying properties across different levels, as illustrated in Fig. 1c.

Table 5: More granular information that better reflects the characteristics of the constructed datasets and distribution shifts.

| 1) **Pileup Shift** — the average number of tracks | | | |
|---|---|---|---|
| Domain | PU-10 | PU-50 | PU-90 |
| #Tracks | 55.76 | 232.58 | 408.38 |
| 2) **Signal Shift** — the average radius of signal tracks | | | |
| Domain | $\tau \to 3\mu$ | $z'_{10} \to 2\mu$ | $z'_{20} \to 2\mu$ | $z \to 2\mu$ |
| #Radius | 3979.66 | 8754.34 | 16014.98 | 58092.27 |
| 3) **Size Shift** — the average number of atoms | | | |
| Domain | Domain-8 | Domain-37 | Domain-95 | Domain-157 |
| #Atoms | 25 | 46 | 105 | 276 |
| 4) **Fidelity Shift** — the average bandgap value | | | |
| Domain | PBE | HLE17 | HSE06* | HSE06 |
| #Bandgap | 2.09 | 2.68 | 2.95 | 3.86 |

## C.2 TRACK DATASET

Here we employ the term *event* to refer to the comprehensive recording of an entire physics process by an experiment (Shlomi et al., 2020). As mentioned in Sec. 3.2.1, a signal event (labeled as *positive*) involves the existence of a particular decay of interest (*i.e.*, signal). Here we are interested in multiple types of signals, including $z \to \mu\mu$, $\tau \to \mu\mu\mu$ (which have been widely observed), and $z'_K \to \mu\mu$ (which is a theoretical possibility) decays. This motivates us to construct the signal shift, where we expect the models trained with multiple types of signals to generalize to new signals that are different but to some extent related to the seen types. *Invariant mass* is a crucial physical quantity that characterizes the distinct decay type. Specifically, when ranked from the largest to the smallest, $z \to \mu\mu$ has an invariant mass of 91.19 GeV, $z'_K \to \mu\mu$ (where we consider $K = 80, 70, 60, 50$ for model training and $K = 10, 20$ for evaluation of model generalizability) has an invariant mass of $K$ GeV, and $\tau \to \mu\mu\mu$ has an invariant mass of 1.777 GeV. In our study, the disparities in invariant mass manifest through changes in the momenta of the signal particles and the radii of signal tracks (tracks left by signal particles). In the $z \to \mu\mu$ decay, the expected radius of the signal tracks is significantly larger, making it easily distinguishable from the background tracks, while in the $\tau \to \mu\mu\mu$ decay, the expected radius of the signal tracks is very close to that of the background tracks.

All events are simulated using the PYTHIA generator (Bierlich et al., 2022) with the addition of soft QCD pileup events, and particle tracks are generated using Acts (Ai et al., 2021). Each point in a data entry is associated with a 3D coordinate, as well as other physical quantities measured by detectors, such as momenta. However, we use a dummy feature with all ones as the point feature for model training, following Miao et al. (2022). The model takes 3D coordinates and the dummy features of each point in data as input and predicts the existence of the signal in the given data.

## C.3 QMOF DATASET

We obtain 3D coordinates of each point in the materials data via the DFT-optimized structures provided by the QMOF Database. For point features, we associate each point in a sample with a categorical feature indicating the atom type for model training. The model takes 3D coordinates and atom-type categorical features as input and predicts the band gap value of given materials data.

Table 6: The criteria of domain splits and sub-group splits in each distribution shift scenario. The "$\mathcal{S}$-Component" and "$\mathcal{T}$-Component" columns provide a description of the composition of the data in the domains $\mathcal{S}$ and $\mathcal{T}$. We denote the number of sub-group splits in the source domain $\mathcal{S}$ as |Sub-groups|. The criterion of the sub-group splits for each scenario is also summarized in the "Criterion" column.

| Dataset | Shift | Shift Case | $\mathcal{S}$-Component | \|Sub-groups\| | Criterion | $\mathcal{T}$-Component |
|---|---|---|---|---|---|---|
| Track | Pileup | PU50 | PU10 | 5 | The number of points | PU50 |
| | | PU90 | | | | PU90 |
| | Signal | $\tau \to 3\mu$ | Mixed Signals: $z \to 2\mu$ and $z'_K \to 2\mu$, where $K = 80, 70, 60, 50$, 5 types in total | 5 | The signal type | $\tau \to 3\mu$ |
| | | $z'_{10} \to 2\mu$ | | | | $z'_{10} \to 2\mu$ |
| | | $z'_{20} \to 2\mu$ | | | | $z'_{20} \to 2\mu$ |
| QMOF | Fidelity | HSE06 | Mixed Fidelity: PBE, HLE17, HSE06* | 3 | The fidelity level | HSE06 |
| | | HSE06* | Mixed Fidelity: PBE, HLE17, HSE06 | 3 | | HSE06* |
| DrugOOD-3D | Assay | lbap-core-ic50-assay | Following DrugOOD | 307 | The assay environment | Following DrugOOD |
| | Size | lbap-core-ic50-size | | 91 | The molecular size | |
| | Scaffold | lbap-core-ic50-scaffold | | 6682 | The scaffold pattern | |

## C.4 DRUGOOD-3D DATASET

We first present how we adapt DrugOOD (Ji et al., 2022) and perform the GDL tasks over the dataset. We pre-process the SMILES (Weininger, 1988) string of data provided in the dataset via the RDKit package (Landrum et al., 2013), generating a conformer for each molecule, so as to assign each atom with a 3D coordinate. Concretely, we begin with generating a molecular object based on the SMILES string. Then we add hydrogens to the molecule and employ the ETKDG method (Riniker & Landrum, 2015) to obtain the initial conformer, which is further refined using the MMFF94 force field (Halgren, 1999). Note that we drop a data entry if it fails in conformer generation after the above process. The model takes 3D coordinates and atom-type categorical features as input, which is analogous to the scenario of the QMOF dataset, and predicts the binding affinity values of given ligands in a form of the binary classification task, as mentioned in Sec. 3.2.3.

For newly created datasets, namely Track-Pileup and Track-Signal, we've got permission from the HEP community and utilized Acts to create them. Acts is licensed under the Mozilla Public License Version 2.0. Others are collected from public datasets and can be found at QMOF and DrugOOD.

# D  DETAILS OF ALGORITHMS AND BACKBONES

## D.1  BACKBONE DETAILS

Our benchmark contains **3** backbones which have been widely used in scenarios of geometric deep learning. Here we give detailed descriptions for each backbone in this study as follows.

- **DGCNN** (Dynamic Graph CNN), introduced by Wang et al. (2019), is a GDL architecture aimed at exploiting local geometric structures of geometric data while maintaining permutation invariance. Specifically, it constructs a local neighborhood graph and applies edge convolution, with dynamic graph updates after each layer of the network.
- **Point Transformer** (Zhao et al., 2021) is an architecture applying self-attention networks to 3D point cloud processing. It is built based on a highly expressive Point Transformer layer, which is invariant to permutation and cardinality of geometric data.
- **EGNN** ($E(n)$ Equivariant Graph Neural Networks), proposed by Satorras et al. (2021), is an architecture that preserves equivariance to rotations, translations and reflections on the coordinates of points when handling GDL data, *i.e.*, $E(n)$ equivariance, and that also preserves equivariance to permutations on the set of points.

## D.2  ALGORITHM DETAILS

Our benchmark contains **11** baselines spanning the No-Info, O-Feature, and Par-Label levels. We group them according to their distinct learning strategies and provide detailed descriptions for each

algorithm as follows. We use ● to represent algorithms from the No-Info level, † for O-Feature, and ‡ for the Par-Label level, respectively.

- *Vanilla*: The empirical risk minimization (ERM) (Vapnik, 1999) minimizes the sum of errors across all samples.

- *Subgroup robustness*: Group distributionally robust optimization (GroupDRO) (Sagawa et al., 2019) aims to minimize worst-case losses and capture subgroup robustness by increasing the importance of groups with larger errors.

- *Invariant learning*: Variance Risk Extrapolation (VREx) (Krueger et al., 2021) captures group invariance by specifically minimizing the risk variances of training domains.

- *Augmentation*: Mixup (Zhang et al., 2018) improves model generalization by linearly interpolating two training samples randomly drawn from the training distribution. We follow Wang et al. (2021) to perform Mixup specifically in the embedding space for the classification of geometric data.

- *Causal Inference*: DIR (Wu et al., 2021) captures the causal rationales for graph-structured data, mainly by conducting interventional augmentation on training data to create multiple interventional distributions, and then filtering out the parts of data that are unstable for model predictions.

- *Information bottleneck*: LRI (Miao et al., 2022) is a novel geometric deep learning strategy grounded on a variational objective derived from the principle of information bottleneck. It injects learnable randomness to each node of geometric data, aimed at capturing minimal sufficient information to make correct and stable predictions. We adopt its *LRI-Bernoulli* framework, which specifically injects Bernoulli randomness to each point.

† *Domain Invariance for Unsupervised Domain Adaptation*: Domain-Adversarial Neural Network (DANN) (Ganin et al., 2016) encourages feature representations to be consistent across the source and the target domain by adversarially training the normal label predictor and a special domain classifier; Deep correlation alignment (DeepCoral) (Sun & Saenko, 2016) also encourages domain invariance by penalizing the deviation of covariance matrices between the source and the target domain.

‡ *Vanilla Fine-tuning*: We fine-tune all parameters of the GDL model using a small amount of OOD data, after it has been pre-trained on ID data via the ERM algorithm. Specifically, we conduct 3 baselines here, fine-tuning the model using 100, 500, and 1000 labeled target samples, respectively.

## E    DETAILS OF EXPERIMENTAL IMPLEMENTATION

We conduct experiments on **3** scientific datasets and **10** cases of distribution shifts, covering **3** GDL backbones and **11** baselines from **3** knowledge levels. We implement our codes based on PyTorch Geometric (Fey & Lenssen, 2019). We provide details of experimental implementation as follows.

**Basic Setup**. For all the experiments, we use the Adam optimizer, with a learning rate of 1e-3 and a weight decay of 1e-5. For each backbone, we use a fixed setting across various scenarios, all with the sum global pooling and the RELU activation function. The settings of batch size, maximum number of epochs, and the number of iterations per epoch for the O-Feature level are consistent across different algorithms for a fair comparison in this study. Details are shown in Table 7. Note that the batch size is set to 128 instead of 256 in the O-Feature level of the pileup shift due to the memory constraints, and maximum number of epochs is set to 75 in the pileup shift because the model has been trained to converge under this setting.

Table 7: General hyperparameters of the datasets in this study.

| Dataset | Shift | No-Info | | O-Feature | | | Par-Label | |
|---|---|---|---|---|---|---|---|---|
| | | Batch Size | # Max Epochs | Batch Size | # Max Epochs | # Iterations per Epoch | Batch Size | # Max Epochs |
| Track | Pileup | 256 | 200 | 128 | 200 | 150 | 256 | 75 |
| | Signal | 256 | 100 | 256 | 100 | 150 | 256 | 100 |
| QMOF | Fidelity | 256 | 100 | 256 | 100 | 150 | 256 | 100 |
| DrugOOD-3D | Size, Scaffold, and Assay | 256 | 100 | 256 | 100 | 150 | 256 | 100 |

**Hyperparameter Tuning**. For each knowledge level and each algorithm, we search from a set of one specific hyperparameter to tune, and select the optimal one based on Val-OOD metric scores for a fair comparison. For VREx, we tune the weight of its variance penalty loss; For GroupDRO, we tune the Exponential coefficient; For Mixup, we tune the probability value that a certain batch data performs mixup augmentation; For DIR, we tune the causal ratio for selecting causal edges; For LRI, we tune the weight of the KL divergence regularizer; For DANN, we tune the weight of the domain classification loss; For DeepCoral, we tune the weight of covariance penalty loss. We detail the search space for each hyperparameter in Table 8.

Table 8: Hyperparameters search space for all algorithms.

| Algorithm | Hyperparameter | Search Space |
|---|---|---|
| VREx | Penalty Weight | $\{0.001, 0.01, 0.1, 1.0\}$ |
| GroupDRO | Exponential Coefficient | $\{0.001, 0.01, 0.1, 1.0\}$ |
| Mixup | Probability | $\{0.25, 0.5, 0.75, 1.0\}$ |
| DIR | Causal Ratio | $\{0.3, 0.4, 0.5\}$ |
| LRI | Information Loss Coefficient | $\{0.01, 0.1, 1.0, 10.0\}$ |
| DANN | Domain Loss Weight | $\{0.001, 0.01, 0.1, 1.0, 5.0\}$ |
| DeepCoral | Penalty Weight | $\{0.001, 0.01, 0.1\}$ |

# F  COMPLETE EXPERIMENTAL RESULTS

Here we present complementary baseline results that are not shown in the main text due to space in Table 9, 10 11, and 12.

Table 9: Experimental results (Val-ID, Test-ID, Val-OOD, and Test-OOD performance included) on the $z'_{20} \rightarrow 2\mu$ case of the Signal shift over three backbones with the evaluation metrics of ACC (higher values indicate better performance). Note that the ID performance of TL methods is not evaluated. Parentheses show standard deviation across 3 replicates. We **bold** and underline the best and the second-best OOD performance for each distribution shift scenario.

| Level | Algorithm | EGNN | | | | DGCNN | | | | Pointtrans | | | |
|---|---|---|---|---|---|---|---|---|---|---|---|---|---|
| | | Val-ID | Test-ID | Val-OOD | Test-OOD | Val-ID | Test-ID | Val-OOD | Test-OOD | Val-ID | Test-ID | Val-OOD | Test-OOD |
| No-Info | ERM | 97.66(0.11) | 96.74(0.18) | 89.41(0.57) | 89.13(0.62) | 96.38(0.12) | 95.55(0.10) | 84.89(0.09) | 84.72(0.30) | 94.60(0.31) | 93.44(0.37) | 81.76(0.61) | 82.73(0.21) |
| | VREx | 97.57(0.12) | 96.75(0.24) | 88.93(0.71) | 89.02(0.71) | 96.16(0.16) | 95.43(0.44) | 84.22(0.29) | 84.17(0.12) | 94.18(0.10) | 93.18(0.34) | 82.54(0.03) | 83.42(0.41) |
| | GroupDRO | 97.62(0.13) | 96.74(0.23) | 89.80(0.37) | **89.55(0.08)** | 95.99(0.34) | 95.12(0.41) | 86.20(0.84) | 85.80(0.92) | 94.41(0.45) | 93.22(0.37) | 83.53(0.74) | 83.90(0.56) |
| | DIR | 94.57(1.81) | 94.12(1.88) | 84.57(2.44) | 84.86(2.62) | 92.98(1.18) | 91.78(0.95) | 83.78(1.25) | 83.98(1.08) | 85.84(10.21) | 84.92(9.97) | 76.39(4.88) | 77.34(5.29) |
| | LRI | 96.96(0.08) | 96.25(0.16) | 86.12(1.07) | 86.49(1.13) | 94.18(0.08) | 93.32(0.07) | 82.28(0.09) | 82.06(0.04) | 92.85(0.18) | 91.75(0.13) | 80.58(0.83) | 81.21(0.41) |
| | MixUp | 97.76(0.04) | 97.05(0.24) | 89.18(0.16) | 88.86(0.41) | 96.48(0.14) | 95.43(0.12) | 85.28(0.58) | 85.15(0.88) | 94.44(0.08) | 93.31(0.05) | 81.91(0.67) | 82.83(0.79) |
| O-Feature | DANN | 96.13(0.20) | 95.16(0.68) | 89.54(0.31) | 89.49(0.31) | 94.99(0.48) | 94.26(0.19) | 88.51(0.21) | **88.33(0.20)** | 90.81(0.26) | 90.13(0.16) | 82.98(0.57) | 83.15(0.50) |
| | Coral | 97.71(0.21) | 96.92(0.20) | 88.86(0.01) | 89.07(0.26) | 95.28(0.11) | 94.54(0.17) | 84.33(0.50) | 84.54(0.59) | 94.17(0.18) | 93.23(0.11) | 81.58(1.23) | 82.50(0.82) |
| Par-Label | TL$_{100}$ | | | 87.28(0.71) | 87.02(1.17) | | | 73.45(0.98) | 71.13(1.25) | | | 82.80(0.80) | 82.98(0.86) |
| | TL$_{500}$ | | | 88.57(0.10) | 87.97(0.23) | | | 79.02(1.52) | 78.31(1.83) | | | 83.68(0.23) | 84.08(0.38) |
| | TL$_{1000}$ | | | 89.55(0.08) | 89.08(0.26) | | | 80.67(1.15) | 79.80(1.52) | | | 84.42(0.38) | **84.59(0.40)** |

Table 10: Experimental results on the **Assay** shift over three backbones, with the evaluation metrics of AUC (higher values indicate better performance). Note that the ID performance of TL methods is not evaluated. Parentheses show standard deviation across 3 replicates. We **bold** and underline the best and the second-best OOD performance for each distribution shift scenario.

| Level | Algorithm | EGNN | | | | DGCNN | | | | Pointtrans | | | |
|---|---|---|---|---|---|---|---|---|---|---|---|---|---|
| | | Val-ID | Test-ID | Val-OOD | Test-OOD | Val-ID | Test-ID | Val-OOD | Test-OOD | Val-ID | Test-ID | Val-OOD | Test-OOD |
| No-Info | ERM | 92.35(0.07) | 91.70(0.11) | 70.66(0.03) | 70.85(0.65) | 90.49(0.06) | 90.07(0.03) | 71.44(0.33) | 70.77(0.34) | 89.54(0.12) | 89.31(0.07) | 70.55(0.36) | 69.58(0.42) |
| | VREx | 92.08(0.09) | 91.67(0.13) | 72.59(1.05) | 71.21(0.47) | 89.81(0.23) | 89.38(0.09) | 71.40(0.16) | 70.72(0.20) | 89.53(0.09) | 89.23(0.07) | 70.25(0.25) | 69.85(0.38) |
| | GroupDRO | 92.05(0.10) | 91.21(0.04) | 72.15(0.24) | **71.82(0.71)** | 88.79(0.16) | 88.45(0.24) | 71.62(0.06) | **71.69(0.70)** | 87.93(0.15) | 87.87(0.15) | 69.94(0.16) | 70.37(0.30) |
| | DIR | 82.57(1.66) | 81.85(1.94) | 70.08(0.98) | 67.97(2.15) | 84.25(0.57) | 83.89(0.52) | 69.91(0.41) | 68.55(1.24) | 86.14(1.09) | 85.99(1.05) | 68.79(0.40) | 68.20(0.23) |
| | LRI | 92.20(0.07) | 91.31(0.15) | 71.31(0.58) | 70.41(0.13) | 90.67(0.09) | 90.13(0.09) | 71.03(0.09) | 70.93(0.34) | 89.28(0.08) | 89.11(0.18) | 69.80(0.11) | 69.83(0.50) |
| | MixUp | 92.25(0.14) | 91.55(0.22) | 71.15(0.18) | 71.36(0.21) | 90.53(0.01) | 90.06(0.12) | 70.88(0.33) | 70.71(0.24) | 89.46(0.14) | 89.20(0.10) | 70.00(0.25) | **70.65(0.41)** |
| O-Feature | DANN | 91.00(0.09) | 90.39(0.07) | 72.13(1.47) | 71.76(0.87) | 90.56(0.13) | 90.28(0.16) | 70.47(0.29) | 70.31(0.42) | 89.65(0.21) | 89.33(0.13) | 69.90(0.22) | 69.71(0.17) |
| | Coral | 92.38(0.05) | 91.84(0.25) | 71.51(0.66) | 71.29(0.55) | 90.59(0.16) | 90.03(0.17) | 70.14(0.97) | 70.80(0.55) | 89.66(0.21) | 89.39(0.15) | 70.02(0.47) | 69.51(0.68) |
| Par-Label | TL$_{100}$ | | | 68.73(0.98) | 68.82(0.47) | | | 67.27(0.53) | 69.14(0.74) | | | 69.31(0.53) | 69.77(0.14) |
| | TL$_{500}$ | | | 70.41(0.30) | 70.81(0.70) | | | 69.01(0.51) | 69.83(0.63) | | | 69.70(0.47) | 70.02(0.28) |
| | TL$_{1000}$ | | | 73.66(1.18) | 71.44(0.49) | | | 70.95(0.53) | 71.19(0.34) | | | 69.61(0.59) | 70.30(0.14) |

## G    COMPLEMENTARY COMPARISON WITH PREVIOUS FINDINGS

Here we conduct comparison between our observations and conclusions with previous findings related to OOD (such as in CV tasks).

### G.1    COMPARISON — CONSISTENCY

- In Sec. 4.2, we observe that fine-tuning can sometimes result in negative effects when the labeled OOD data is quite limited, particularly in cases involving a smaller degree of distribution shifts. This is consistent with (Kirkpatrick et al., 2017), where fine-tuning a large model based on a small set of labels may lead to catastrophic forgetting. To mitigate this issue, the strategy of surgical fine-tuning raised in (Lee et al., 2022), *i.e.*, fine-tuning a limited, contiguous subset of all pre-trained model layers, is a potential solution for this problem.

- In Sec. 4.2, we observe that multiple OOD generalization methods in our No-Info level find it hard to provide significant improvement across various applications. And we can find consistent observations in existing works (Gulrajani & Lopez-Paz, 2020; Koh et al., 2021).

- In Sec. 4.3, we conclude that, "For the OOD generalization methods to learn robust representations, the more informatively the groups obtained by splitting the source domain $S$ indicate the distribution shift, the better performance these methods may achieve." This is consistent with previous CV works (Creager et al., 2021), which revealed the importance of appropriate subgroup partitioning for invariant learning.

- Some works focus on leveraging additional auxiliary variables for OOD generalization. Xie et al. (2020) used auxiliary information to help improve OOD performance in a semi-supervised scenario. Lin et al. (2022) recently proposed to leverage such additional variables to encode information about the latent distribution shift, and to jointly learn group splits and invariant representation. How to leverage these auxiliary variables to enhance OOD generalization is an interesting topic for GDL.

### G.2    COMPARISON — DISPARITY

Firstly, the second point in Sec. G.1 could be even more severe in GDL compared to CV tasks considering the intricate nature of irregularity and geometric prior (information on the structure space and symmetry properties like invariance or equivariance) inherent in geometric data.

Moreover, certain shifts in scientific GDL are infrequent or even unique in CV. This indicates the challenges faced by several methods initially proposed for CV tasks in addressing these shifts, and the necessity to develop OOD methods specifically designed for scientific GDL. Here are some examples in our work.

- Size shift, despite categorized as covariate shift, is a unique case where the model trained in data with lower size is to generalize to data with larger size. Methods designed for CV might struggle to capture this mechanism, potentially explaining why several methods do not perform well in the context of size shift in our study.

- Fidelity shift, which indicates the change of causal correlation between $X_c$ and Y, corresponds to a significant challenge in material property prediction. However, most methods in our benchmark find it hard to handle such shift, except the pertaining-finetuning strategy.

## H    COMPLEMENTARY ANALYSIS OF EXPERIMENTAL RESULTS

### H.1    FURTHER ILLUSTRATIONS OF THE FIDELITY SHIFT

We use Kernel Density Estimation (KDE) (Rosenblatt, 1956; Parzen, 1962) to estimate the marginal label distribution $\mathbb{P}(Y)$ in the fidelity and assay shifts. As shown by Fig. 3a and 3b, in the HSE06 case, there is a significant disparity between $\mathbb{P}_{\mathcal{S}}(Y)$ and $\mathbb{P}_{\mathcal{T}}(Y)$, whereas in the HSE06* scenario, such difference between $\mathbb{P}_{\mathcal{S}}(Y)$ and $\mathbb{P}_{\mathcal{T}}(Y)$ is much smaller. This corroborates the analysis presented in Sec. 4.3, highlighting that the TL strategies yield greater benefits when the concept shift exhibits a substantial disparity in $\mathbb{P}(Y)$ between the domains $\mathcal{S}$ and $\mathcal{T}$.

Table 11: Experimental results (Val-ID and Val-OOD performance) on **Pileup** (PU50 and PU90 cases), **Signal** ($\tau \to 3\mu$ and $z'_{10} \to 2\mu$ cases), **Size**, **Scaffold**, and **Fidelity** (HSE06 and HSE06* cases) shifts over the backbones of EGNN and DGCNN. Note that Val-ID performance of TL methods is not evaluated. Parentheses show standard deviation across 3 replicates. ↑ denotes higher values correspond to better performance, whereas ↓ denotes lower for better. We **bold** and underline the best and the second-best OOD performance for each distribution shift scenario.

**Pileup Shift — $\mathcal{I}$-Conditional Shift (ACC↑)**

| | | EGNN | | | | DGCNN | | | |
| --- | --- | --- | --- | --- | --- | --- | --- | --- | --- |
| | | PU50 | | PU90 | | PU50 | | PU90 | |
| Level | Algorithm | Val-ID | Val-OOD | Val-ID | Val-OOD | Val-ID | Val-OOD | Val-ID | Val-OOD |
| | ERM | 96.18(0.05) | 88.68(0.31) | 96.29(0.22) | 82.76(0.88) | 94.66(0.43) | 86.49(1.10) | 94.66(0.43) | 79.63(1.48) |
| | VREx | 96.05(0.12) | 88.36(0.43) | 96.05(0.12) | 81.63(0.73) | 94.95(0.22) | **86.92(0.59)** | 94.95(0.22) | **80.65(0.93)** |
| No-Info | GroupDRO | 93.09(0.36) | 83.82(0.33) | 93.09(0.36) | 76.71(0.19) | 91.79(0.16) | 79.94(0.36) | 91.79(0.16) | 74.45(0.21) |
| | DIR | 95.50(0.19) | 86.57(0.40) | 95.50(0.19) | 79.74(0.31) | 94.44(0.18) | 84.48(0.46) | 94.44(0.18) | 76.43(1.19) |
| | LRI | 96.28(0.09) | 88.88(0.27) | 95.89(0.23) | 82.80(0.71) | 94.52(0.13) | 86.08(0.09) | 94.52(0.13) | 79.61(0.62) |
| | MixUp | 96.25(0.10) | **89.29(0.24)** | 96.25(0.10) | 82.67(0.41) | 94.86(0.38) | 86.29(0.46) | 94.86(0.38) | 80.42(0.66) |
| O-Feature | DANN | 95.53(0.39) | 87.60(0.86) | 95.96(0.11) | 82.16(0.83) | 94.35(0.21) | 85.29(0.56) | 94.69(0.43) | 76.87(1.69) |
| | Coral | 95.69(0.23) | 87.65(0.87) | 95.88(0.20) | 79.73(2.43) | 94.46(0.28) | 84.97(1.04) | 94.91(0.42) | 78.24(1.08) |
| | TL$_{100}$ | | 85.79(0.53) | | 80.31(0.70) | | 81.79(0.94) | | 74.13(0.38) |
| Par-Label | TL$_{500}$ | | 87.76(0.27) | | 83.31(0.96) | | 84.91(1.09) | | 79.00(1.17) |
| | TL$_{1000}$ | | 88.57(0.19) | | **84.87(0.53)** | | 85.19(0.86) | | 79.94(0.20) |

**Signal Shift — $\mathcal{C}$-Conditional Shift (ACC↑)**

| | | EGNN | | | | DGCNN | | | |
| --- | --- | --- | --- | --- | --- | --- | --- | --- | --- |
| | | $\tau \to 3\mu$ | | $z'_{10} \to 2\mu$ | | $\tau \to 3\mu$ | | $z'_{10} \to 2\mu$ | |
| Level | Algorithm | Val-ID | Val-OOD | Val-ID | Val-OOD | Val-ID | Val-OOD | Val-ID | Val-OOD |
| | ERM | 97.85(0.12) | 67.11(0.49) | 97.72(0.04) | 71.30(0.78) | 96.55(0.15) | 66.12(0.41) | 96.47(0.07) | 69.71(0.20) |
| | VREx | 97.71(0.18) | 67.51(0.66) | 97.51(0.12) | 72.08(0.52) | 96.30(0.14) | 66.13(0.19) | 96.42(0.22) | 69.53(0.55) |
| No-Info | GroupDRO | 97.38(0.17) | 67.92(0.17) | 97.68(0.24) | 73.25(0.34) | 95.81(0.17) | 67.20(0.42) | 95.75(0.24) | 71.15(0.38) |
| | DIR | 77.80(2.78) | 68.47(0.13) | 93.94(2.86) | 71.18(0.91) | 92.91(0.47) | 66.21(0.64) | 92.91(0.47) | 71.45(0.42) |
| | LRI | 96.96(0.08) | 68.34(0.28) | 96.95(0.12) | 70.76(0.72) | 91.31(0.69) | 68.59(0.08) | 94.32(0.38) | 68.91(0.08) |
| | MixUp | 97.74(0.30) | 66.55(1.30) | 97.83(0.08) | 71.37(1.24) | 96.25(0.16) | 66.23(1.22) | 96.48(0.22) | 69.99(0.28) |
| O-Feature | DANN | 82.06(1.10) | **69.45(0.12)** | 91.20(1.04) | **77.15(0.45)** | 81.37(1.60) | 69.17(0.02) | 88.97(0.29) | **75.61(0.16)** |
| | Coral | 96.94(0.76) | 67.96(0.65) | 97.70(0.08) | 71.97(0.26) | 95.52(0.08) | 65.91(0.76) | 95.17(0.15) | 69.88(0.76) |
| | TL$_{100}$ | | 64.15(1.10) | | 74.26(0.48) | | 65.57(0.80) | | 65.07(1.10) |
| Par-Label | TL$_{500}$ | | 68.35(0.33) | | 75.97(0.49) | | 67.75(0.93) | | 67.73(0.63) |
| | TL$_{1000}$ | | 69.04(0.14) | | 76.97(0.21) | | 68.56(0.10) | | 68.47(0.86) |

**Size & Scaffold Shift — Covariate Shift (AUC↑)**

| | | EGNN | | | | DGCNN | | | |
| --- | --- | --- | --- | --- | --- | --- | --- | --- | --- |
| | | Size | | Scaffold | | Size | | Scaffold | |
| Level | Algorithm | Val-ID | Val-OOD | Val-ID | Val-OOD | Val-ID | Val-OOD | Val-ID | Val-OOD |
| | ERM | 91.83(0.21) | 78.96(0.07) | 94.16(0.19) | 75.89(0.78) | 90.32(0.03) | 77.04(0.10) | 91.16(0.10) | 75.98(0.22) |
| | VREx | 91.56(0.23) | 78.94(0.42) | 94.41(0.36) | 76.41(1.04) | 90.07(0.12) | 77.47(0.27) | 91.85(0.74) | 76.63(0.47) |
| No-Info | GroupDRO | 87.46(0.28) | 74.08(0.27) | 94.22(0.26) | 76.77(0.66) | 83.99(0.24) | 73.05(0.12) | 91.56(0.19) | **76.79(0.13)** |
| | DIR | 87.83(1.03) | 75.57(0.46) | 89.23(2.45) | 73.46(2.61) | 80.99(0.32) | 72.03(0.63) | 79.66(1.19) | 73.09(0.80) |
| | LRI | 91.85(0.22) | **79.24(0.29)** | 94.35(0.22) | 76.38(0.09) | 90.27(0.38) | 77.23(0.14) | 87.83(0.16) | 75.82(0.31) |
| | MixUp | 91.70(0.27) | 79.05(0.23) | 94.09(0.24) | 77.32(0.15) | 90.24(0.14) | 77.35(0.24) | 91.90(0.09) | **76.81(0.32)** |
| O-Feature | DANN | 91.98(0.15) | 79.07(0.12) | 94.88(0.12) | 76.65(0.14) | 89.79(0.17) | 77.04(0.17) | 91.59(0.51) | 75.70(0.26) |
| | Coral | 92.07(0.20) | 79.01(0.44) | 95.15(0.18) | 76.81(0.29) | 89.68(0.20) | **77.62(0.22)** | 91.89(0.50) | 75.40(0.25) |
| | TL$_{100}$ | | 77.53(0.69) | | 74.90(0.70) | | 76.49(0.26) | | 74.99(0.57) |
| Par-Label | TL$_{500}$ | | 77.80(0.50) | | 77.12(0.83) | | 76.59(0.22) | | 76.18(0.27) |
| | TL$_{1000}$ | | 77.99(0.18) | | **77.64(0.26)** | | 76.57(0.11) | | 76.45(0.27) |

**Fidelity Shift — Concept Shift (MAE↓)**

| | | EGNN | | | | DGCNN | | | |
| --- | --- | --- | --- | --- | --- | --- | --- | --- | --- |
| | | HSE06 | | HSE06* | | HSE06 | | HSE06* | |
| Level | Algorithm | Val-ID | Val-OOD | Val-ID | Val-OOD | Val-ID | Val-OOD | Val-ID | Val-OOD |
| | ERM | 0.498(0.006) | 1.128(0.094) | 0.618(0.005) | 0.541(0.007) | 0.486(0.005) | 1.126(0.032) | 0.601(0.004) | 0.537(0.009) |
| No-Info | VREx | 0.498(0.005) | 1.110(0.068) | 0.619(0.004) | **0.524(0.013)** | 0.508(0.003) | 1.060(0.089) | 0.619(0.003) | 0.520(0.009) |
| | GroupDRO | 0.530(0.000) | 1.029(0.029) | 0.674(0.003) | **0.525(0.004)** | 0.512(0.005) | 1.012(0.017) | 0.684(0.007) | **0.505(0.003)** |
| O-Feature | DANN | 0.495(0.002) | 1.185(0.017) | 0.620(0.001) | 0.542(0.010) | 0.484(0.004) | 1.093(0.033) | 0.603(0.003) | 0.534(0.007) |
| | Coral | 0.499(0.007) | 1.182(0.044) | 0.618(0.005) | 0.554(0.006) | 0.489(0.001) | 1.100(0.017) | 0.603(0.002) | 0.526(0.010) |
| | TL$_{100}$ | | 0.726(0.010) | | 0.606(0.028) | | 0.702(0.019) | | 0.583(0.003) |
| Par-Label | TL$_{500}$ | | 0.640(0.008) | | 0.543(0.016) | | 0.625(0.005) | | 0.524(0.006) |
| | TL$_{1000}$ | | **0.619(0.007)** | | 0.535(0.012) | | **0.586(0.004)** | | 0.508(0.002) |

## H.2 COMPLEMENTARY ANALYSIS OF THE ASSAY SHIFT

Apart from the fidelity shift, we have an intriguing discovery in the assay shift which is also closely linked to the analysis we have presented in Sec. 4.3. As illustrated in Fig. 3c and 3d, although there is a large divergence of $\mathbb{P}(Y)$ between different assay subgroups, the distribution $\mathbb{P}(Y)$ is quite

Table 12: Experimental results (Val-ID, Test-ID, Val-OOD, and Test-OOD performance included) on **Pileup** (PU50 and PU90 cases), **Signal** ($\tau \to 3\mu$ and $z'_{10} \to 2\mu$ cases), **Size**, **Scaffold**, and **Fidelity** (HSE06 and HSE06* cases) shifts over the backbone of **Point Transformer**. Note that the ID performance of TL methods is not evaluated. Parentheses show standard deviation across 3 replicates. ↑ denotes higher values correspond to better performance, whereas ↓ denotes lower for better. We **bold** and underline the best and the second-best OOD performance for each distribution shift scenario.

**Pileup Shift — $\mathcal{I}$-Conditional Shift (ACC↑)**

| Level | Algorithm | PU50 | | | | PU90 | | | |
|---|---|---|---|---|---|---|---|---|---|
| | | Val-ID | Test-ID | Val-OOD | Test-OOD | Val-ID | Test-ID | Val-OOD | Test-OOD |
| No-Info | ERM | 93.93(0.36) | 93.15(0.31) | 85.25(0.25) | 84.07(0.60) | 93.93(0.36) | 93.15(0.31) | 79.73(0.10) | 78.67(0.25) |
| | VREx | 93.74(0.42) | 93.17(0.33) | 84.95(0.66) | 83.75(0.29) | 93.74(0.42) | 93.17(0.33) | 79.41(0.39) | 77.92(0.19) |
| | GroupDRO | 92.27(0.30) | 91.59(0.21) | 82.49(0.75) | 81.35(0.78) | 92.27(0.30) | 91.59(0.21) | 74.45(1.52) | 73.66(1.49) |
| | DIR | 93.15(0.13) | 92.81(0.14) | 84.79(0.64) | 84.13(0.45) | 93.15(0.13) | 92.81(0.14) | 79.71(0.83) | 78.92(0.76) |
| | LRI | 93.58(0.20) | 92.96(0.27) | 83.93(0.27) | 83.63(0.63) | 93.58(0.20) | 92.96(0.27) | 78.77(0.41) | 77.77(0.50) |
| | MixUp | 93.79(0.12) | 93.16(0.24) | 85.41(0.24) | **84.55(0.59)** | 93.79(0.12) | 93.16(0.24) | 80.17(0.34) | **79.15(0.46)** |
| O-Feature | DANN | 93.82(0.13) | 93.01(0.14) | 85.06(0.34) | 84.25(0.60) | 93.75(0.21) | 92.84(0.31) | 78.22(0.81) | 77.11(0.89) |
| | Coral | 93.45(0.07) | 92.88(0.14) | 84.87(0.10) | 83.97(0.12) | 93.59(0.12) | 92.88(0.12) | 77.51(1.20) | 76.38(1.73) |
| Par-Label | TL$_{100}$ | | | 83.39(0.38) | 82.76(0.30) | | | 77.95(1.62) | 76.59(1.88) |
| | TL$_{500}$ | | | 84.59(0.07) | 83.54(0.45) | | | 79.19(0.43) | 78.28(0.70) |
| | TL$_{1000}$ | | | 84.82(0.18) | 84.32(0.27) | | | 79.89(0.13) | 78.64(0.33) |

**Signal Shift — $\mathcal{C}$-Conditional Shift (ACC↑)**

| Level | Algorithm | $\tau \to 3\mu$ | | | | $z'_{10} \to 2\mu$ | | | |
|---|---|---|---|---|---|---|---|---|---|
| | | Val-ID | Test-ID | Val-OOD | Test-OOD | Val-ID | Test-ID | Val-OOD | Test-OOD |
| No-Info | ERM | 94.60(0.31) | 93.44(0.37) | 68.66(0.14) | 67.43(0.13) | 94.60(0.31) | 93.44(0.37) | 68.25(0.39) | 66.96(0.92) |
| | VREx | 94.29(0.25) | 93.24(0.15) | 68.68(0.21) | 67.63(0.08) | 94.29(0.25) | 93.24(0.15) | 69.88(0.23) | 68.28(0.40) |
| | GroupDRO | 94.22(0.40) | 92.98(0.52) | 67.65(0.32) | 66.46(0.43) | 94.22(0.40) | 92.98(0.52) | 70.60(0.61) | 69.15(0.89) |
| | DIR | 85.84(10.21) | 84.92(9.97) | 68.16(0.55) | 67.31(0.49) | 85.84(10.21) | 84.92(9.97) | 68.59(1.93) | 66.64(1.81) |
| | LRI | 92.85(0.18) | 91.75(0.13) | 68.74(0.02) | 67.55(0.08) | 92.85(0.18) | 91.75(0.13) | 69.56(0.30) | 68.22(0.89) |
| | MixUp | 94.51(0.10) | 93.47(0.24) | 68.63(0.16) | 67.58(0.06) | 94.44(0.08) | 93.31(0.05) | 69.07(0.88) | 67.41(1.29) |
| O-Feature | DANN | 94.53(0.30) | 93.20(0.14) | 68.69(0.04) | **67.64(0.09)** | 83.97(0.30) | 84.07(0.35) | 72.26(0.10) | **70.75(0.35)** |
| | Coral | 94.42(0.24) | 93.32(0.25) | 68.71(0.05) | 67.60(0.05) | 94.61(0.27) | 93.44(0.29) | 68.39(0.98) | 67.47(0.92) |
| Par-Label | TL$_{100}$ | | | 66.39(1.66) | 65.87(1.20) | | | 69.57(1.63) | 68.92(1.96) |
| | TL$_{500}$ | | | 68.52(0.26) | 67.60(0.23) | | | 68.80(1.21) | 67.69(1.08) |
| | TL$_{1000}$ | | | 68.63(0.13) | 67.32(0.10) | | | 70.82(2.03) | 69.81(2.00) |

**Size & Scaffold Shift — Covariate Shift (AUC↑)**

| Level | Algorithm | Size | | | | Scaffold | | | |
|---|---|---|---|---|---|---|---|---|---|
| | | Val-ID | Test-ID | Val-OOD | Test-OOD | Val-ID | Test-ID | Val-OOD | Test-OOD |
| No-Info | ERM | 88.91(0.28) | 88.09(0.58) | 76.34(0.25) | 64.17(0.49) | 90.05(0.25) | 81.22(0.42) | 75.26(0.61) | 67.92(0.46) |
| | VREx | 88.44(0.28) | 87.90(0.35) | 76.30(0.16) | 64.44(0.34) | 89.35(0.31) | 80.96(0.33) | 75.20(0.19) | 67.97(0.47) |
| | GroupDRO | 83.52(0.20) | 82.71(0.37) | 71.89(0.37) | 58.19(0.46) | 89.29(0.67) | 81.05(0.29) | 75.32(0.25) | 67.93(0.27) |
| | DIR | 83.65(2.49) | 83.46(2.40) | 73.63(1.14) | 62.82(0.91) | 83.61(3.02) | 77.05(1.05) | 72.11(0.67) | 65.82(1.11) |
| | LRI | 88.34(0.58) | 87.70(0.77) | 76.35(0.24) | 64.43(0.45) | 85.70(0.27) | 79.08(0.21) | 74.15(0.18) | 67.34(0.15) |
| | MixUp | 88.76(0.07) | 88.17(0.21) | 76.58(0.13) | 63.81(0.13) | 89.40(0.46) | 80.88(0.22) | 75.00(0.07) | 67.56(0.20) |
| O-Feature | DANN | 88.13(0.12) | 87.61(0.07) | 76.12(0.16) | 64.76(0.33) | 89.87(0.16) | 80.70(0.20) | 74.30(0.24) | 67.26(0.31) |
| | Coral | 88.33(0.70) | 87.91(0.38) | 76.60(0.15) | 64.57(0.12) | 90.26(0.47) | 80.44(0.40) | 74.49(0.69) | 67.45(0.36) |
| Par-Label | TL$_{100}$ | | | 75.90(0.20) | 64.11(0.38) | | | 74.07(0.56) | 67.67(0.09) |
| | TL$_{500}$ | | | 75.97(0.37) | 64.33(0.47) | | | 75.20(0.52) | 68.35(0.18) |
| | TL$_{1000}$ | | | 75.89(0.36) | **65.14(0.90)** | | | 76.32(0.41) | **70.00(0.15)** |

**Fidelity Shift — Concept Shift (MAE↓)**

| Level | Algorithm | HSE06 | | | | HSE06* | | | |
|---|---|---|---|---|---|---|---|---|---|
| | | Val-ID | Test-ID | Val-OOD | Test-OOD | Val-ID | Test-ID | Val-OOD | Test-OOD |
| No-Info | ERM | 0.492(0.002) | 0.495(0.002) | 1.182(0.014) | 1.146(0.014) | 0.613(0.003) | 0.624(0.007) | 0.543(0.002) | 0.553(0.003) |
| | VREx | 0.522(0.009) | 0.517(0.008) | 1.102(0.044) | 1.080(0.033) | 0.621(0.003) | 0.623(0.004) | 0.523(0.002) | 0.536(0.004) |
| | GroupDRO | 0.527(0.005) | 0.516(0.008) | 0.993(0.052) | 0.959(0.057) | 0.641(0.015) | 0.643(0.016) | 0.513(0.004) | 0.529(0.003) |
| O-Feature | DANN | 0.493(0.001) | 0.501(0.003) | 1.162(0.033) | 1.135(0.038) | 0.612(0.001) | 0.615(0.004) | 0.537(0.007) | 0.560(0.011) |
| | Coral | 0.491(0.003) | 0.498(0.000) | 1.212(0.027) | 1.181(0.033) | 0.612(0.006) | 0.618(0.013) | 0.541(0.006) | 0.561(0.003) |
| Par-Label | TL$_{100}$ | | | 0.684(0.010) | 0.689(0.015) | | | 0.583(0.008) | 0.598(0.006) |
| | TL$_{500}$ | | | 0.618(0.008) | 0.613(0.005) | | | 0.519(0.005) | 0.545(0.006) |
| | TL$_{1000}$ | | | 0.583(0.001) | **0.584(0.002)** | | | 0.511(0.007) | **0.522(0.010)** |

similar between the source and target domain, *i.e.* $\mathbb{P}_{\mathcal{S}}(Y) \approx \mathbb{P}_{\mathcal{T}}(Y)$, which stands in contrast to the scenarios of the scaffold shift and size shift shown in Fig. 3e and 3f. This provides a plausible explanation for why TL strategies cannot exhibit substantial improvement over ERM in the assay shift, as shown in Table 10, although it is also categorized as the concept shift. However, to provide a comprehensive answer to this question, it's crucial to consider various other factors as well. For example, we follow DrugOOD to transition the affinity prediction to a binary classification task, which is a kind of simplification made for this problem; Besides, the mechanism of the assay shift,

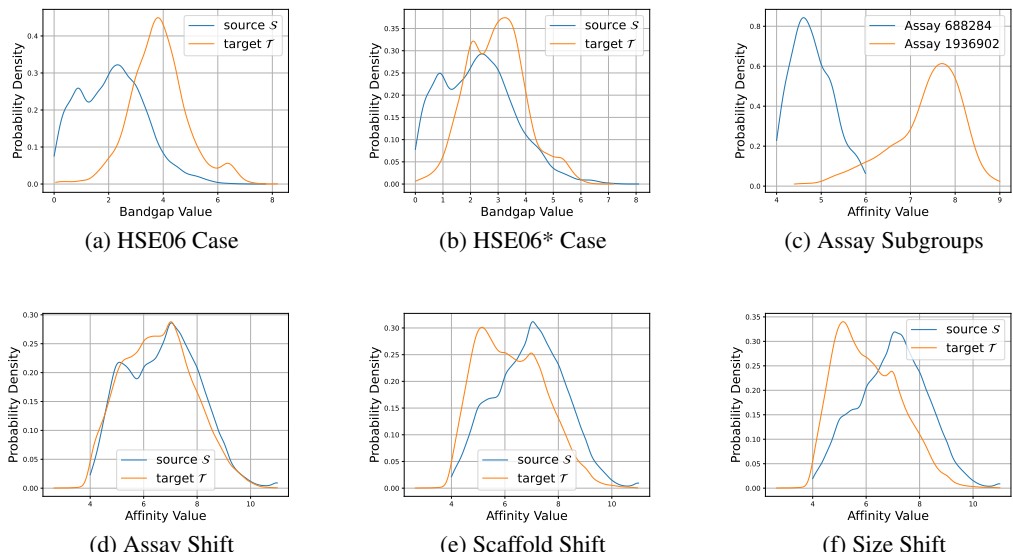

Figure 3: Plotted KDE curves of the marginal label distribution $\mathbb{P}(Y)$. (a) / (b): $\mathbb{P}_{\mathcal{S}}(Y)$ and $\mathbb{P}_{\mathcal{T}}(Y)$ in the HSE06 / HSE06* case of the fidelity shift, respectively, where $Y \in \mathcal{Y}$ represents the ground-truth band gap value; (c): $\mathbb{P}(Y)$ between two distinct assay subgroups, namely Assay 688284 and Assay 1936902, where $Y$ represents the ground-truth binding affinity value; (d) / (e) / (f): $\mathbb{P}_{\mathcal{S}}(Y)$ and $\mathbb{P}_{\mathcal{T}}(Y)$ in the assay / scaffold / size shift.

unlike the fidelity shift scenario, may go beyond a mere label distribution mismatch but could involve more aspects, like the substantial shift in the input distribution $\mathbb{P}(X)$ between the domains $\mathcal{S}$ and $\mathcal{T}$, which can pose a significant challenge, particularly when the amount of labeled OOD data is limited. We leave further comprehensive analysis to future work.

