# OpenReview forum: "GDL-DS: A Benchmark for Geometric Deep Learning under Distribution Shifts"
_ICLR.cc/2024/Conference — Submitted to ICLR 2024_

### Official Review · Reviewer_1MFG · 2023-10-29

**Soundness:** 4 excellent
**Presentation:** 4 excellent
**Contribution:** 4 excellent
**Rating:** 8
**Confidence:** 4

**Summary:**

The paper addresses the challenge of distribution shifts in Geometric Deep Learning (GDL), a topic that has seen limited research focus despite GDL's prominence in various scientific applications. The authors introduce GDL-DS, a comprehensive benchmark designed to evaluate the performance of GDL models across scenarios that encounter distribution shifts. They provide a comprehensive evolution on several datasets from different fields; particle physics, materials science, and biochemistry, and categorize distribution shifts into three types: conditional, covariate, and concept shifts. Furthermore, they explore three levels of out-of-distribution (OOD) information access and evaluate multiple GDL backbones and learning algorithms. The benchmark consists of 30 experiment settings, and the findings provide valuable insights for researchers and practitioners in the GDL domain.

**Strengths:**

- The paper presents a comprehensive benchmark for GDL models, covering a spectrum of scientific domains and distribution shifts. Such a benchmark fills an existing gap in the literature.

- The authors leverage the causality inherent in scientific applications to classify distribution shifts into conditional, covariate, and concept shifts, providing a clearer understanding of the challenges faced.

- By exploring three distinct levels of OOD information, the paper offers a nuanced understanding of the impact of OOD data on model performance, addressing disparities in previous works.

- The paper conducts a myriad of experiments, with 30 different settings, evaluating various GDL backbones and learning algorithms, ensuring a robust and holistic evaluation.

- The results yield key takeaways that can guide the selection of practical solutions based on the availability of OOD data, serving as a valuable resource for researchers and practitioners.

**Weaknesses:**

Given the disparities in previous benchmarking studies across various domains,, there's a compelling case to expand this benchmark study to encompass both CV and NLP tasks to provide a holistic and unified perspective on performances across diverse tasks.

**Questions:**

- How do the findings of this study compare with earlier research on CV and NLP tasks concerning distribution shifts?

- What is the rationale behind the choice of the considered GL backbones? Would incorporating more diverse GDL backbones or learning algorithms significantly alter the conclusions drawn from this study?

---

> ### Author Response · Authors · 2023-11-17
>
> We express our sincere gratitude to reviewer 1MFG for reviewing our paper, appreciating our work and raising valuable comments.  We address your questions as follows.
>
>   **1). How do the findings of this study compare with earlier research on CV and NLP tasks concerning distribution shifts?**
>
> Firstly, we’d like to appreciate earlier OOD works on CV and NLP tasks, many of which have shed valuable insights into our research. We have meticulously connected our multiple conclusions with the existing OOD work in CV and NLP and put details in the appendix. The details are as follows.
>
> - In Sec. 4.2, we observe that fine-tuning can sometimes result in negative effects when the labeled OOD data is quite limited. **This is consistent with catastrophic forgetting in [1]. To mitigate this issue, the strategy of surgical fine-tuning raised in [2]** is a potential solution for this problem.
>
> - In Sec. 4.2, we observe that multiple OOD generalization methods in our No-Info level find it hard to provide significant improvement across various applications. We can find **similar observations in existing CV/NLP works, including [3], [4]**. Actually, this could be even **more severe in GDL compared to CV tasks**, considering the intricate nature of irregularity and geometric prior (information on the structure space and symmetry properties like invariance or equivariance) inherent in geometric data.
>
> - In Sec. 4.3, we conclude that “For the OOD generalization methods to learn robust representations, the more informatively the groups obtained by splitting the source domain $S$ indicate the distribution shift, the better performance these methods may achieve.” This is **consistent with previous CV works like [5]**, which revealed the importance of appropriate subgroup partitioning for invariant learning. And we note that **[6] proposed to leverage** additional auxiliary variables for enhancing OOD generalization. This may also shed insights into future research on scientific GDL.
>
> Moreover, certain shifts in scientific GDL are different from common shifts in CV/NLP, which leads to challenges when addressing these shifts with methods initially proposed for CV/NLP tasks. Here are some examples in our work:
>
> - Shifts related to graph/geometric structure: For instance, the I-conditional shift can be interpreted as a size shift (the number of nodes in data), where the model trained in data with smaller sizes is to generalize to data with larger sizes. We observed that the methods designed for CV struggle to perform well. In addition, the features in GDL additionally incorporate the geometric structure. The methods in CV work on feature representations without explicitly handling the change in geometric structure, which may result in suboptimal solutions.
>
> - Fidelity shift, which indicates the change of causal correlation between $X_c$ and Y, corresponds to a significant challenge in material property prediction. However, we rarely encounter the shift in the causal relationship in CV, so most methods in our benchmark find it hard to handle such a shift, except the pertaining-finetuning strategy.
>
> The above two points of **difference indicate the necessity to develop OOD methods specifically designed for GDL**.
> Overall, We thank the reviewer’s suggestion and the comparisons made above would enhance the connection between our findings with earlier research in CV/NLP domains and shed more insights for future research.
>
> [1] Kirkpatrick, James, et al. "Overcoming catastrophic forgetting in neural networks." Proceedings of the national academy of sciences 114.13 (2017): 3521-3526.
>
> [2] Lee, Yoonho, et al. "Surgical Fine-Tuning Improves Adaptation to Distribution Shifts." ICLR, 2022.
>
> [3] Gulrajani, Ishaan, and David Lopez-Paz. "In Search of Lost Domain Generalization." ICLR, 2020.
>
> [4] Koh, Pang Wei, et al. "Wilds: A benchmark of in-the-wild distribution shifts." ICML, 2021.
>
> [5] Creager, Elliot, Jörn-Henrik Jacobsen, and Richard Zemel. "Environment inference for invariant learning." ICML, 2021.
>
> [6] Lin, Yong, et al. "ZIN: When and How to Learn Invariance Without Environment Partition?." NeurIPS, 2022.

---

> > ### Author Response · Authors · 2023-11-17
> >
> > **2).	What is the rationale behind the choice of the considered GDL backbones? Would incorporating more diverse GDL backbones or learning algorithms significantly alter the conclusions drawn from this study?**
> >
> > We consider choosing the three GDL backbones from the following 2 aspects.
> >
> > Firstly, they are representative in different categories of GDL backbones. In our selected backbones, DGCNN is an early work of 3D-CNN designed for processing point cloud data; Point Transformer is a Transformer-based GDL architecture; EGNN is an E(n)-Equivariant Graph Neural Networks.
> >
> > Secondly, they have been widely used for various scenarios of scientific GDL, such as [7], [8], and [9]. We are aware that there are several wonderful works regarding GDL network architecture designed for some specific scientific task, but we did not choose them considering that they are not easily applicable to diverse scientific domains.
> >
> > Overall, we found that the order of the results is basically consistent in different backbones, and our chosen baselines tend to span a variety of learning mechanisms, so we believe that the overall conclusions would not be significantly altered by adding more backbones or algorithms.
> >
> > [7] Qu, Huilin, and Loukas Gouskos. "Jet tagging via particle clouds." Physical Review D 101.5 (2020): 056019.
> >
> > [8] Atz, Kenneth, Francesca Grisoni, and Gisbert Schneider. "Geometric deep learning on molecular representations." Nature Machine Intelligence 3.12 (2021): 1023-1032.
> >
> > [9] Gagliardi, Luca, et al. "SHREC 2022: protein–ligand binding site recognition." Computers & Graphics 107 (2022): 20-31.

---

> > > ### Comment · Reviewer_1MFG · 2023-11-22
> > >
> > > Thanks to the authors for their comprehensive and detailed response to the concerns I raised. Following a thoughtful review of your response, I have concluded to keep my initial score unchanged. Your efforts in addressing these issues are greatly appreciated and have contributed meaningfully to the review process.

---

> > > > ### Author Response · Authors · 2023-11-23
> > > > **Thanks**
> > > >
> > > > Many thanks for checking our response. Thank you for your kind words and support!
> > > >
> > > > the authors

---

### Official Review · Reviewer_GGxS · 2023-10-31

**Soundness:** 2 fair
**Presentation:** 2 fair
**Contribution:** 2 fair
**Rating:** 3
**Confidence:** 5

**Summary:**

The paper presents a OOD benchmark for geometric deep learning in science. The authors curate datasets from 3 scientific domains, identify several shifts in each dataset, and conduct 3 OOD splits for each shift. Then each setting is used to evaluate 3 GDL backbones and several OOD methods.

**Strengths:**

1. The paper focuses a a very compelling topic. OOD datasets and benchmarks for geometric deep learning in science are innovative and meaningful research.

2. The paper presentation includes rich contents, with tables and figures well organized.

3. The selected data presents practical tasks. The conducted experiments look correct and sufficient experimental analyses are given.

**Weaknesses:**

1. The use of critical terms should be better considered. Concept drift is a well-established term in the study of causality and distribution shift. As defined in [1], which the authors also cited, the only constraint for concept drift is "changes in $p(y|X)$". To avoid any confusions to readers, this conventional definition should be followed without modifications like $P(Y|X_c)$. Similarly for the definition of covariate shift. If the authors attempt to define a more specific kind of shift, another term should be used.

2. The causal statements are problematic.
    - The statement that $X$ consists of two disjoint parts and $X_i ⊥Y |X_c$ does not hold. A easy violation would be $X_c →Y →X_i$. Intrinsically, $Y$ is often a property of the input and therefore $X$ cannot be divided into two disjoint parts that are causal/independent, but there would exist a part of $X$ that is statistically associated with $Y$ while non-causal to $Y$. A classic example is the PIIF and FIIF causal modeling, such as the analysis in [2].
    - Following the above point, even for $X → Y$, $P(Y|X)P(X) = P(Y|X_c)P(X)$ does not hold. For $X → Y$, there can be a case where $P_S(Y |X_i)\neq P_T (Y |X_i)$, which will also result in a "conditional shift". It is also included by the definition of concept shift. Constraining $Y → X$ does not seem like a necessity for conditional shift.
    - Overall, as the foundation of the whole paper, 3.1 appears to be logically unclear and farraginous and needs major corrections.

3. Contribution overclaimed and related works not well addressed. In the comparison with existing benchmarks, the authors claim no existing OOD benchmarks consider conditional shift, which is not true. OoD-Bench, GDS, and GOOD all include the Cmnist dataset, which is clearly conditional shift. GOOD also constructed conditional shift for each dataset. Also, though benchmarks like WILDS do not use test labeled/unlabeled data for algorithm learning, these OOD info are available. Therefore, Table 1 gas multiple overclaiming issues, and the authors should treat existing works properly.

4. Experimental setting not fair. Some methods are trained solely on the Train-ID dataset, while DA algorithms are trained on both Train-ID and OOD input data, and TLs also learn labeled Train-OOD data. This does not seem like a fair setting since different methods are trained on even different numbers of data samples. Given that the analyses are conducted based on comparing all these methods together, a fair evaluation setting is certainly needed.

5. Baselines out-of-date. These years many new OOD methods including new sota have been proposed. The benchmark should include more recent methods as baselines. For learning algorithms the sota methods on the Wilds leaderboard should be considered. For graph OOD methods, many recent methods can easily outperform DIR. Also, geometric methods specifically developed for scientific tasks should be considered.

6. The benchmark includes only 3 datasets. Though more than one shift is identified for each dataset, this number seems a bit few for a benchmark. Given that the datasets are not newly collected, possibly more discussions on contributions like curating 3D coordinate can make up for the overall contribution.


[1] A survey on concept drift adaptation

[2] Invariant risk minimization

**Questions:**

See Weaknesses

**Details Of Ethics Concerns:**

The license for each dataset is not addressed in the paper.

---

> ### Author Response · Authors · 2023-11-17
>
> We thank reviewer GGxS for recognizing the compelling aspect of our work in developing OOD benchmarks for GDL in science, well-organized contents, and sufficient experimental analyses. Based on the review, we acknowledge that the reviewer has read our paper thoroughly and pointed out questions, in particular identifying our improper underemphasis of some previous works’ coverage. However, we respectfully disagree with the reviewer’s comments on the soundness of our assumption of categorizing various distribution shifts and concerns regarding our experiment settings. We summarize the reviewer’s concerns and provide our detailed explanations in the following response.
>
>   **1). Use of terms: “The use of critical terms should be better considered.”**
>
> We thank the reviewer for raising this confusion based on our paper presentation. We are not aiming to modify the existing well-established terms. Instead, we would like to further explain and extend the common definition of these well-established terms under the context of our selected application datasets. We have clarified in our refined draft by first following the well-established definitions of these distribution shifts and then highlighting that we further extend the definition to what we present in the paper given our assumption.
>
> **2).	Soundness of causal statements in section 3.1**
>
> Overall, the reviewer mentioned some alternatively correct definitions of causal relationships that are not covered by this paper. However, the existence of those alternative definitions does not imply the lack of soundness of our ways of definitions. Note that our definitions in Sec. 3.1 are to model the scientific GDL datasets benchmarked in this work, which indicate how these datasets were curated and how their distribution shifts are like. Our definitions are not to cover all possible causality relationships. The alternative definitions mentioned by the reviewer, though existing in theory, either are not reflected by those datasets or may introduce more confusion. We are sorry for not clarifying this point in Sec. 3.1, which caused such confusion. In the following, we respond to each bullet point raised by the reviewer and have adjusted the exposition in Sec. 3.1 accordingly to avoid the similar confusion made by the reviewer.
>
> - 2.1 	soundness of causal statements: “The statement that $X$ consists of two disjoint parts and $X_i \bot Y|X_c $ does not hold”
>
> This review confuses “the settings this work is targeting” and “other possible settings that this work is not to cover”. Note that such conditional independence just describes the settings approximately indicated by our datasets. We agree that there exist other types of dependence, while they are out of the scope of this work because we did not find interesting scientific GDL datasets that reflect those types of dependence. Moreover, limiting the scope properly allows for more in-depth study without losing much generality.
>
> First, such conditional independence is derived from real-world scientific application and holds in our established scenarios. Take our track Dataset in particle physics as an example. The entire point cloud can be split into 2 parts, namely the signal part used to determine whether there is a specific signal we are looking for (determine the label), and the background part, which itself is independent of the label given the signal part. This exactly follows the conditional independence assumption as $X_i \bot Y|X_c$. The other two datasets also comply with the assumption that the label is determined by only a part of the data entry, with another part being statistically correlated but conditionally independent of the label given the causal part. Therefore, we respectfully disagree with the argument that “X cannot be divided into two disjoint parts that are causal/independent”, which is at least properly reflected by the datasets considered in this work.
>
> Second, this setting of conditional independence is of a broad range of research interest to the machine learning community. For instance, we can find similar formulations in DIR [1], Invariant Rationalization [2], etc. Take DIR for example, Equation 1 in this paper indicates the conditional-independence property of the oracle rationale, where $S$ is “statistically associated with $Y$ but non-causal to $Y$”, and is independent of $Y$ given the causal part $C$.
>
> In a word, the example $X_c\to Y\to X_i$ raised by the reviewer exists, but it is beyond the scope of this work, especially when we did not find a scientific GDL dataset that obeyed such a setting. We are happy to enrich our benchmark in the future if the reviewer may suggest some scientific GDL datasets that reflect the settings the reviewer would like to cover.
>
> [1] Wu, Yingxin, et al. "Discovering Invariant Rationales for Graph Neural Networks." ICML, 2021.
>
> [2] Chang, Shiyu, et al. "Invariant rationalization." ICML, 2020.

---

> > ### Author Response · Authors · 2023-11-17
> >
> > Continued from 2) above
> >
> > - 2.2	Involving data generation process when characterizing shift: “Constraining $Y\to X$ does not seem like a necessity for conditional shift”
> >
> > We understand the reviewer’s comment that the data generation process $X \to Y$ or $Y \to X$ is not a necessity to characterize a type of distribution shift. However, given a data generation process ($X \to Y$ or $Y \to X$), it becomes more clear to tell by domain experts which type of distribution shift a dataset satisfies. For instance, pairing the generating process $X\to Y$ with $P_S(Y|X_c)\ne P_T(Y|X_c)$ perfectly describes the mechanism of fidelity shift, an important real-world scenario in molecular dynamic simulations in materials science. And plus pairing $Y\to X$ with $P_S(X|Y)\ne P_T(X|Y)$ perfectly aligns with the proposed scenarios of particle physics, as $Y$ indicates the type of particle decay while $X$ is the particle hit data generated from this type of decay.
> >
> > As the reviewer argued, the data generation process $X \to Y$ may also satisfy conditional shift $P_S(X|Y)\ne P_T(X|Y)$ (in theory), but it may also satisfy $P_S(X|Y)=P_T(X|Y)$. There is not a simple way to tell whether the conditional shift exists and how large such the conditional shift could be. There is possibly $P_S(Y|X_i)\ne P_T(Y|X_i)$ that can also derive $P_S(X|Y)\ne P_T(X|Y)$ but the datasets considered in this work do not have $P_S(Y|X_i)\ne P_T(Y|X_i)$. Instead, for the data generated via process $Y \to X$, it is easy to tell the existence of $P_S(X|Y)\ne P_T(X|Y)$ and the degree of such a shift by analyzing how $X$ is generated given $Y$.  Our way of categorization is more succinct and aligns well with practical understandings.
> >
> > Moreover, the seminal work in ML [3] that initiated the study of conditional shift also incorporates the data generation process $Y\to X$. Our work follows this tradition.
> >
> > [3] Zhang, Kun, et al. "Domain adaptation under target and conditional shift." ICML, 2013.
> >
> > **3).	Contribution overclaimed**
> > - 3.1	“CMNIST dataset in OoD-Bench, GDS, and GOOD is clearly conditional shift.”
> >
> > The reviewer raises that the Cmnist Dataset follows a conditional shift while we missed discussing it in our literature review. We appreciate the reviewer for this insightful comment . After revisiting the papers, we found they satisfied the concept shift definition as a change in $P(Y|X)$. When constructing Table 1, we didn’t mark them as “concept shift” since their mechanism is different from what we formalized as the “concept shift” in our work. We are sorry for being too specific in Table 1 and missed this point. In the revised manuscript, we revised Table 1 and clarified the missing contributions of previous works. In the following response, we will detail how the shift in CMNIST correlates to our work.
> >
> > All these works mentioned in this comment did not explicitly define the shift as “conditional shift”. From the explicit formulations established in OoD-Bench[4] and GOOD[5], it corresponds to concept shift, characterized by the change of $P(Y|X)$. So we tend to treat it as “concept shift” rather than “conditional shift” to avoid confusion.
> > Also, it is important to note that this concept shift in GOOD has **a different mechanism than the one formalized in our study**: In our work, concept shift particularly denotes the change in causal correlation between $X_c$ and $Y$. In contrast, in GOOD's context, concept shift corresponds to the change in statistical rather than causal correlation, such as correlations between color and digit in CMNIST.
> >
> > Overall, we have added more comparison & clarification in the refined draft and tend not to make confusion. We thank the reviewer again for pointing out this.
> >
> > - 3.2	OOD info is available in WILDS
> >
> > We thank the reviewer for pointing out the confusion. In Table 1, the column “Available OOD Info” actually means how previous work leveraged the learning algorithms corresponding to the three OOD-Info levels. WILDS did not use test labeled/unlabeled data for algorithm learning.
> > We have made further clarification in Table 1 in the refined draft.
> >
> > [4] Ye, Nanyang, et al. "Ood-bench: Quantifying and understanding two dimensions of out-of-distribution generalization." CVPR, 2022.
> >
> > [5] Gui, Shurui, et al. "Good: A graph out-of-distribution benchmark." NeurIPS, 2022.
> >
> > [6] Wiles, Olivia, et al. "A Fine-Grained Analysis on Distribution Shift." ICLR, 2021.

---

> > > ### Author Response · Authors · 2023-11-17
> > >
> > > **4). The unfairness of the model training process**
> > >
> > > This comment proposed by the reviewer precisely pertains to a key aspect of our contributions & motivation, while unfortunately causing some confusion on the reviewer’s side. We are sorry for the confusion and here are some clarifications.
> > >
> > > We **intentionally** incorporate different levels of OOD information into the training process to investigate how training with different OOD information helps with model generalization. Comparing OOD methods stand-alone is not our goal. Instead, we are to compare different combinations of OOD methods with a certain level of OOD data info.  Such comparisons uncover useful insights into handling scientific application OOD challenges: The insights can guide researchers in selecting the appropriate category of methodology when having different levels of available OOD information under a specific shift. Researchers may even decide whether it is valuable to collect some unlabeled or labeled data from the testing domain by trading off between the cost of collecting such data and the expected performance gain given such extra OOD data info told by this work. Moreover, we claim we are fair in the evaluation process by fairly evaluating the boost of model generalization across the three settings on the same validation and testing dataset, which is well addressed in Sec. 4.1 of our paper.
> > >
> > > **5). Other concerns**
> > > - 5.1	Baselines out of date
> > >
> > > Firstly, the goal of the paper is not to find the SOTA methods. We select algorithms that are well-known and span a broad range of learning strategies under different levels of OOD info, including 1) vanilla ERM, 2) invariant learning, 3) data augmentation, 4) subgroup robustness, 5) causal inference, 6) information bottleneck, 7) domain invariance and adaptation (adversarial training & distance regularization), 8) Pre-training and Fine-tuning. Our selection of algorithms includes various fundamental learning principles in the OOD field, and newly proposed methods are also generally based on the aforementioned principles.
> > >
> > > We are happy to include more methods to compare if those methods introduce principles to deal with OOD challenges that have not been covered by the above already included principles.
> > >
> > > - 5.2	Lack in the number and originality of datasets
> > >
> > > We respectfully argue that the review substantially underestimates our contribution. We have 3 scientific domains and 6 datasets in total, including Track-Pileup, Track-Signal, QMOF, DrugOOD-3D-Assay, DrugOOD-3D-Size, DrugOOD-3D-Scaffold, which can be seen in Table 2. We respectfully disagree with just treating one scientific domain as a dataset, otherwise DrugOOD [7], a wonderful benchmark in AI-aided Drug Discovery, would have had only a single dataset. Moreover, the two datasets in particle physics, i.e, Track-Pileup and Track-Signal, are newly created by ours. They are two fundamentally different application tasks in particle physics, of which each has thousands of researchers focused all over the world.
> > >
> > > - 5.3	Ethics flag: no copyright. “The license for each dataset is not addressed in the paper.”
> > >
> > > We thank the reviewer for pointing this out. For newly created datasets, namely Track-Pileup and Track-Signal, we’ve got permission from the HEP community and utilized Acts to create them. Acts is licensed under the Mozilla Public License Version 2.0 (https://www.mozilla.org/en-US/MPL/2.0/). Other collected datasets are public and can be found at https://github.com/Andrew-S-Rosen/QMOF and https://github.com/tencent-ailab/DrugOOD. We will add them to the refined draft.
> > >
> > > [7] Ji, Yuanfeng, et al. "DrugOOD: Out-of-Distribution (OOD) Dataset Curator and Benchmark for AI-aided Drug Discovery--A Focus on Affinity Prediction Problems with Noise Annotations." arXiv preprint, 2022.

---

> ### Author Response · Authors · 2023-11-23
> **Looking forward to your feedback**
>
> Dear reviewer GGxS,
>
> After carefully considering your comments, we provide point-to-point answers to address all your concerns. As the discussion period is closing, we would appreciate it if you could give some feedback. Your participation is important to  the review process.
>
> the authors

---

> > ### Comment · Reviewer_GGxS · 2023-11-23
> > **Response to rebuttal**
> >
> > Many thanks to the detailed rebuttal from the authors.
> > I have checked and appreciate the response and paper revision. However, most of my concerns are not addressed.
> > The causal modeling and math in 3.1 is still rough with no major changes. The concept shift and conditional shift notions for related works are not clear. The shifts in OoD-Bench, GDS, and GOOD are conceptually closer to the conditional shift defined in this work than concept shift, since $P(Y|X_c)$ does not change at all in those works. This is also why I believe a rigorous causal modeling is vital as the theoretical foundation, considering all the potential misconceptions. Also, an obvious unfairness in experimental settings, as I mentioned, is that comparisons are made given that different numbers of data samples are used to train different methods. How can the influence of OOD info be demonstrated when the number of data samples is not controlled? Finally, I still believe SOTA methods of the field should be considered for newly proposed benchmarks. Given the aforementioned reasons, I have to maintain my rating of this work for now.

---

> > > ### Author Response · Authors · 2023-11-23
> > > **Thanks!**
> > >
> > > Thanks for checking our response. Besides the other two subjective concerns, we do not think our comparison is unfair. Although   methods that use different categories indeed adopt different amount of data to train the model, the methods in the same category adopt the same data for training. Moreover, the labeled in-domain data, the unlabeled out-of-domain data, the labeled out-of-domain data keep unchanged across all methods as long as one method will use any of these three types.
> > >
> > > Again, we greatly appreciate the reviewer's feedback, though we respectfully disagree it.

---

> ### Author Response · Authors · 2023-11-23
> **Official Comment by Authors**
>
> Many thanks to the response of the reviewer. We will also give more explanations in the first concern:
>
> The reviewer comments "The shifts in OoD-Bench, GDS, and GOOD are conceptually closer to the conditional shift". However,
> the original papers formalized the shift by using a concept shift notion: In OoD-bench, the authors formalized $p(y | z) \ne q(y | z)$ and in GOOD, the authors formalized $P ^{train}(Y |X) \ne P ^{test}(Y |X)$. These are concept shifts according to their explicit formulations, and we tried to follow their notions.
>
> Thank you for your response!

---

### Official Review · Reviewer_xYh1 · 2023-11-01

**Soundness:** 2 fair
**Presentation:** 3 good
**Contribution:** 2 fair
**Rating:** 6
**Confidence:** 2

**Summary:**

The paper effectively addresses the challenge of evaluation of deep learning models generalization abilities under distribution shift in geometric deep learning (point cloud data). It categorizes various sources of distribution shift between training and testing domains and introduces a new benchmark dataset spanning three distinct domains: particle collision physics, chemistry, and material science. The paper further evaluates multiple models, drawing conclusions and recommendations regarding which methods generalize better in specific scenarios

**Strengths:**

The introduction of a new benchmark dataset that spans different domains and types of distribution shifts is a noteworthy contribution. This dataset allows for a more nuanced comparison of deep learning methods based on the specific type of shift, making it practically significant and important for the research community.

The paper's coverage of various scientific fields, including particle collision physics, chemistry, and material science, broadens its applicability and relevance, potentially opening up opportunities for interdisciplinary research.

The paper is clearly written and technically sound.

**Weaknesses:**

It's crucial to include detailed information about the characteristics of the new benchmark datasets and of the already existing datasets. Providing information on data size and other characteristics would enhance the reader's understanding of the datasets' properties and its applicability.

**Questions:**

n/a

---

> ### Author Response · Authors · 2023-11-17
>
> We express our gratitude to reviewer xYh1 for acknowledging the contribution of our benchmark and recognizing the clarity and technical soundness of our paper. We also appreciate the valuable advice provided by the reviewer regarding the inclusion of more detailed information about the characteristics of the datasets in our benchmark. Basic dataset information, including data size and strategies for domain/subgroup splits, has been included in Appendix C. In our refined draft, we will provide more granular information that better reflects the characteristics of the constructed datasets and distribution shift. This will involve details covering:
>
> 1) the average number of tracks and particles for each pileup level in Pileup Shift (Track Dataset)
> 2) the average signal radius of each type of signal in Signal Shift (Track Dataset)
> 3) the average number of atoms for ID and OOD Dataset in Size Shift (DrugOOD-3D Dataset)
> 4) the average band gap value for each fidelity level in Fidelity Shift (QMOF Dataset)
>
> We will collect and add the above 4 types of information to the refined draft before the deadline for rebuttal. Feel free to let us know if more details should be provided.

---

> > ### Author Response · Authors · 2023-11-21
> > **Looking forward to your feedback**
> >
> > Dear reviewer xYh1,
> >
> > We have added the promised more details about the dataset. Please let us know if any further details are needed.
> >
> > the authors

---

### Author Response · Authors · 2023-11-17

Dear reviewers,

Many thanks for your time and efforts in providing us with these valuable comments to improve the paper presentation. We are grateful that all reviewers appreciate the contributions of our work in developing a new benchmark for geometric deep learning (GDL) that spans various scientific domains and types of distribution shifts.

In addition, we appreciate the actionable feedback from each reviewer for us to improve our work further. Specifically, reviewer xYh1 suggests more detailed introductions regarding the characteristics of proposed datasets. Reviewer GGxS is mainly concerned with our models to characterize different categories of distribution shifts and some experimental settings. Reviewer 1MFG would like us to compare further with distribution shift issues in CV and NLP. In the following, we will address their comments respectively in our response to each reviewer.

---

### Author Response · Authors · 2023-11-21
**Looking forward to feedback from the reviewers**

Dear reviewers,

We want to draw your attention to our submission. After carefully considering your comments, we provide point-to-point answers to address all your concerns. As the discussion period is closing, we would appreciate it if you could give some feedback.

Thanks for your attention and participation!

Best,

Authors

---

### Meta-Review · Area_Chair_Gvwy · 2023-12-17

**Metareview:**

This work presents a benchmark for comparing geometric deep learning models in out-of-distribution (OOD) data domains. The authors take care to include a number of domains both in terms of datasets used as well as classes of OOD problems. They test a number of algorithms from the literature on these models to demonstrate the benchmarking.

The reviewers for this work were of mixed opinion. On one hand one reviewer considered the work highly relevant and well described and executed. A second reviewer appreciated the work but had concerns on the dataset descriptions and details that lead to a more lukewarm assessment. A third reviewer noted a number of weaknesses and concerns, in particular in clarity when explaining the classes of OOD problems, the lack of SOTA methods benchmarked, and potential fairness issues on running the multiple models. The authors took great effort to respond to these concerns. Some, such as the inclusion of more recent methods, seem to be improved in the revision, however I think that the manuscript can be greatly helped by a more careful rewriting to clearly convey their approach. Specifically in the area of clearly detailing how their running of different data maintains of fair comparison, as well as a careful rewriting of the presentation of different OOD settings. Therefore I do not recommend this work be accepted.

**Justification For Why Not Higher Score:**

There were a number of concerns about clarity and fairness of the benchmarking that could and should be better clarified.

**Justification For Why Not Lower Score:**

N/A

---

### Decision · Program_Chairs · 2024-01-16

Reject